# DeepResearchGym: A Free, Transparent, and Reproducible Sandbox for Deep Research

## Abstract

Deep research systems represent an emerging class of agentic information retrieval methods that generate comprehensive and well-supported reports to complex queries. However, most existing frameworks rely on dynamic commercial search APIs, which pose reproducibility and transparency challenges in addition to their cost. To address these limitations, we introduce DeepResearchGym as an open-source sandbox that combines a reproducible search API with a rigorous evaluation protocol for benchmarking deep research systems. The API indexes large-scale public web corpora, namely ClueWeb22 and FineWeb, using a state-of-the-art dense retriever and approximate nearest neighbor search via DiskANN. It achieves lower latency than popular commercial APIs while ensuring stable document rankings across runs, and is free for research use. To evaluate deep research systems' outputs, we extend the Researchy Questions benchmark with automatic metrics through LLM-as-a-judge to measure alignment with users' information needs, retrieval faithfulness, and report quality. Experimental results show that systems integrated with DeepResearchGym achieve performance comparable to those using commercial APIs, with performance rankings remaining consistent across evaluation metrics. A case study on short-answer search agents further demonstrates the sandbox's utility for cost-effective training, showing that models trained within the sandbox can generalize to commercial search.

## 1 Introduction

Recent advances in Large Language Models (LLMs) have driven a transformation in information access paradigms, moving beyond ranked retrieval toward systems capable of synthesizing comprehensive report-style responses to complex queries. These deep research systems aim to address complex and open-ended information needs, combining iterative retrieval with multi-step reasoning and generation, autonomously navigating and evaluating diverse sources to construct well-supported reports. Prominent commercial examples include OpenAI (OpenAI, 2025) and Perplexity (Perplexity AI, 2025) deep research modes, which have demonstrated how these systems can significantly enhance user experience when addressing intricate questions requiring synthesis across multiple sources. Recent industry developments further underscore this shift, with Google moving towards AI-driven search tools (Reid, 2024), and Apple announcing plans to integrate services such as OpenAI and Perplexity into its Safari browser (Gurman et al., 2025).

As deep research systems are gaining prominence, they also introduce novel evaluation challenges. Being agentic by design, these systems rely on iterative search, retrieval, and reasoning over vast collections of online data, making evaluation dependent on access to environments with diverse coverage that faithfully simulate real-world behavior. Yet, such infrastructures remain scarce to the research community, forcing reliance on commercial web search APIs. While convenient, these APIs introduce critical limitations: their proprietary nature restricts transparency in the retrieval processes, hindering research on search itself, and their continuous evolution undermines reproducibility and fair benchmarking.

To address these challenges, we introduce DEEPRESEARCHGYM as an open-source benchmarking framework specifically designed to enable transparent and reproducible evaluation of deep research systems. At the core of our framework is a free and open-source search API built upon public web snapshots comprising millions of documents, such as ClueWeb22 (Overwijk et al., 2022) and FineWeb (Penedo et al., 2024). This API exposes standardized endpoints for both document retrieval and content access, enabling integration with long-form generation pipelines.

Our search infrastructure design emphasizes transparency and reproducibility, aiming to support realistic search behavior without the variability introduced by commercial services. The retrieval pipeline consists of publicly available components, including the document collections, a state-of-the-art embedding model, and a scalable approximate nearest neighbor search index. This setup allows researchers to audit system behavior, analyze the influence of retrieved evidence, and rerun deep research experiments under reproducible search conditions, since retrieval results remain stable over time. We provide code to support local deployment of DEEPRESEARCHGYM's infrastructure, supporting full pipeline reproducibility, as well as experiments using different retrieval models and/or document collections. Empirical evaluations show that the system achieves strong retrieval quality with minimal loss from approximate search, while maintaining response times below those attained by commercial APIs.

Furthermore, DEEPRESEARCHGYM includes an evaluation protocol designed to assess long-form deep research systems. We build upon the Researchy Questions dataset (Rosset et al., 2024), which was initially created as a retrieval benchmark curated from commercial search logs. This dataset represents high-engagement non-factoid queries, making it a suitable testbed for deep research systems. Our evaluation extension shifts the focus from assessing retrieval effectiveness to evaluating the quality of deep research systems' responses. We employ an LLM-as-a-judge methodology (Gu et al., 2024) to automatically evaluate responses across key dimensions - alignment with users' information needs, factual grounding, and overall report quality - leveraging Researchy Questions' ground-truth documents to yield more reliable judgments.

To empirically ground our framework, we apply DEEPRESEARCHGYM's evaluation protocol to assess a diverse set of commercial and open-source deep research systems. Our findings highlight two key insights: first, systems maintain performance across evaluation metrics when integrated with DEEPRESEARCHGYM's search API, indicating that our infrastructure maintains report quality on par with commercial search setups. Second, comprehensive coverage of user information needs remains the most challenging dimension, indicating room for improvement in how current systems address complex, multi-faceted queries. Beyond evaluation, a case study demonstrates that agents trained within the sandbox generalize to commercial search at inference time, achieving comparable learning gains to commercial-trained agents while avoiding monetary API costs. Together, the results support DEEPRESEARCHGYM as a promising sandbox environment for deep research, validated by approximately 12 million API queries processed during its initial months of public availability.

## 2 RELATED WORK

Early work on Retrieval-Augmented Generation (RAG) systems focused on improving performance on knowledge-intensive question answering by retrieving supporting documents from large corpora and conditioning generation on this evidence to enhance factual accuracy (Lewis et al., 2020; Thakur et al., 2025; Zhou et al., 2024). Building on this foundation, several deep research systems have been optimized for short-form factoid-style answering. These include reinforcement learning approaches that enable search agents to autonomously navigate the web, issue iterative queries, and synthesize concise responses (Jin et al., 2025a; Song et al., 2025; Zheng et al., 2025), as well as prompt-based methods like Search-o1 (Li et al., 2025b), which equips LLMs with the ability to trigger web searches when encountering knowledge gaps, leveraging the collected evidence to guide synthesis. While effective for short-form question answering, these approaches are not designed to support the generation of detailed reports that require broader synthesis, reasoning, and integration across multiple sources (OpenAI, 2025).

A complementary line of work has advanced towards comprehensive long-form report generation frameworks. GPTResearcher (Elovic, 2025) orchestrates agentic workflows to coordinate planning, retrieval, and drafting across hybrid data sources, incorporating techniques such as report planning (Wang et al., 2023) and query decomposition (Bonchi et al., 2008) to enhance long-form synthesis, while enforcing completeness.

Building on these paradigms, other deep research systems emphasize agentic tool use to extend reasoning capabilities beyond pure text-based retrieval (Han et al., 2025; Nguyen et al., 2025). For instance, Open-DeepSearch (Alzubi et al., 2025) implements two agentic variants: one that follows an action-observation cycle, allowing the model to iteratively query external resources and refine its reasoning; and another that augments this by generating and executing Python scripts for more complex computational tasks. Agentic Reasoning (Wu et al., 2025b) similarly combines multi-agent collaboration with code execution, contextual memory, and dynamic knowledge-graph construction via a dedicated mind-map agent, enabling structured exploration of complex problems. HuggingFace's OpenDeepResearch initiative (HuggingFace, 2025) follows similar directions in an open-source framework while emphasizing transparency and modularity. A common limitation across aforementioned systems is their reliance on commercial web search APIs such as Tavily (Tavily, 2025) and SERPer (Serper, 2025) for document retrieval. These APIs provide limited transparency into document indexing and ranking, are subject to changes over time, and restrict researchers' ability to fully replicate retrieval conditions, posing challenges for reproducibility and fair evaluation.

Parallel efforts have also targeted the evaluation of deep research systems' quality. In particular, multiple benchmarks have driven progress on short-form expert question answering, such as GAIA (Mialon et al., 2024), HLE (Phan et al., 2025), and FRAMES (Krishna et al., 2025). Recent work has introduced frameworks that move beyond short-form QA and address the challenges of evaluating long-form synthesis. FACTScore (Min et al., 2023) and SAFE (Wei et al., 2024) decompose outputs into atomic claims and verify their factual consistency against external sources. For retrieval-augmented systems, ARES (Saad-Falcon et al., 2024) and RAGChecker (Ru et al., 2024) offer modular evaluations that explicitly link generated claims to retrieved evidence, providing fine-grained diagnostics of relevance and faithfulness. Long$^2$RAG (Qi et al., 2024) extends this approach by introducing Key Point Recall (KPR), which evaluates how well long-form answers capture essential content from retrieved sources by measuring coverage of salient points.

## 3 DEEPRESEARCHGYM

This section presents DEEPRESEARCHGYM as an open-source framework designed to support reproducible research on deep research systems. To address the challenges related to the reliance on commercial web search APIs, DEEPRESEARCHGYM offers a controlled sandbox environment built on large-scale web corpora. It provides a state-of-the-art retrieval API, and an evaluation protocol to measure long-form report quality.

### 3.1 SEARCH SANDBOX

This subsection introduces our search API, designed to enable reproducible retrieval for deep research systems. We begin by describing the underlying web corpora, followed by an overview of the dense retriever and the ANN indexing approach used to enable efficient search. Finally, we outline the API interface, including available endpoints, supported arguments, and response format.

#### 3.1.1 WEB CORPORA

DEEPRESEARCHGYM indexes three large-scale web datasets, namely the English subsets of ClueWeb22 A and B (Overwijk et al., 2022), and the FineWeb CC-MAIN-2024-51 snapshot (Penedo et al., 2024).

ClueWeb22 was collected in 2022 and comprises approximately 10 billion web pages. It is organized into three categories, each representing different segments of the web. Category B, known as ClueWeb22-B,

approximates the *super head* of the web, encompassing the most frequently visited pages (e.g., pages from Wikipedia, major news outlets, and other top domains). It includes around 200 million web pages, with approximately 87 million in English. These pages were sampled based on their likelihood to satisfy user information needs, as estimated by a commercial search engine's importance scoring. Low-quality and spam pages were filtered during sampling to enhance the dataset's overall quality.

To mitigate potential coverage concerns and ensure that systems can be exposed to a broader spectrum of web content, we also provide access to ClueWeb22-A. This larger subset encompasses approximately 1 billion English pages from the *mostly head* of the web, offering a more diverse mix of frequently visited websites.

FineWeb is a large-scale English web corpus collected from 96 Common Crawl snapshots between 2013 and 2024. It comprises approximately 15 trillion tokens of cleaned and deduplicated web data. The dataset employs rigorous filtering, deduplication, and quality control measures, resulting in a high-quality resource for LLM training. To mitigate temporal constraints associated with ClueWeb22, we focus on the most recent crawl, which includes over 180 million documents capturing more recent trends compared to earlier data. This makes the collection particularly valuable for queries that require up-to-date information, reflecting the evolving nature of web content and user interests.

### 3.1.2 SEARCH INDEXES

To enable efficient state-of-the-art retrieval across our selected corpora, we built a distributed dense retrieval backend combining state-of-the-art embedding models and approximate nearest neighbor search. Specifically, we leverage the `MiniCPM-Embedding-Light` model (Hu et al., 2024; OpenBMB, 2024), i.e. an open-source dense retriever trained on 260 million query-document pairs, generating 1024-dimensional document representations. The model leverages bidirectional attention mechanisms (BehnamGhader et al., 2024) and weighted mean pooling (Muennighoff, 2022) to capture long-range dependencies in documents with up to 8192 tokens. It achieves competitive performance on multiple benchmarks, and shows good generalization ability given a zero-shot performance of 55.27 in nDCG@10 on the BEIR benchmark (Thakur et al., 2021), outperforming other popular alternatives such as `bge-large-en-v1.5` (BAAI, 2024) and `jina-embeddings-v3` (JinaAI, 2024), which achieve 54.29 and 53.88 in nDCG@10, respectively.

We index the document embeddings using DiskANN (Subramanya et al., 2019). Each corpus is partitioned into independent shards of 25 million documents, which are separately indexed for distributed deployment. During search, shards are queried in parallel, and the top-ranked results are merged to produce a final ranking.

To ground the retrieval effectiveness of our search system, we evaluated it on the Researchy Queries test set. This experiment used ClueWeb22-B as the corpus, since the Researchy Queries relevance labels are grounded on it. Table 1 presents the retrieval performance, considering the number of retrieved documents $K = 100$, while varying $L$, i.e. a DiskANN search-time parameter that controls the size of the candidate neighbor list explored during search. Increasing $L$ typically boosts recall and ranking quality by allowing more thorough exploration of the search graph, but comes at the cost of reduced query throughput. We provide metrics computed given the ground-truth clicked documents (MRR@n, nDCG@n, and R@n), as well as approximate nearest neighbor recall (ANN R@n), computed based on exact-search results. The results with increased $L$ indicate that the error introduced by ANN search is minimal, solidifying the retrieval quality.

### 3.1.3 RETRIEVAL API

DEEPRESEARCHGYM provides a retrieval API designed to support deep research systems over the aforementioned corpora. The API exposes two primary endpoints: (i) the `/search` endpoint, which accepts a text query and returns a ranked list of documents from the selected corpus, and (ii) the `/fetch` endpoint, which retrieves the archived textual content of a document given its URL. For both endpoints, users can choose which corpus to use (i.e., ClueWeb-B, ClueWeb-A, or FineWeb) through an API parameter.

Table 1: Retrieval performance of the DEEPRE-SEARCHGYM /search API as measured over the Researchy Questions test set.

| | Relevance Eval | | | ANNS Eval | |
|---|---|---|---|---|---|
| $L$ | MRR@10 | nDCG@10 | R@100 | R@10 | R@100 |
| 100 | 48.34 | 39.40 | 78.06 | 90.01 | 88.72 |
| 200 | 48.39 | 39.49 | 78.27 | 92.63 | 91.01 |
| 300 | 48.41 | 39.50 | 78.35 | 93.87 | 92.64 |
| 400 | 48.44 | 39.52 | 78.39 | 94.72 | 93.68 |
| 500 | 48.45 | 39.55 | 78.43 | 95.39 | 94.39 |

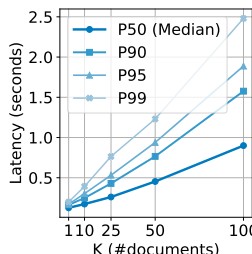 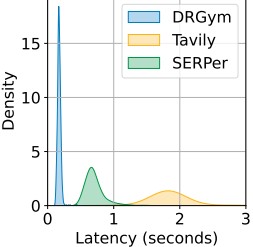

Figure 1: Latency percentiles with varying $K$ for DEEPRESEARCHGYM (left), and latency comparison with commercial APIs for $K = 10$ (right).

The /search endpoint supports document retrieval over the previously introduced corpora, i.e. ClueWeb22 and FineWeb. By operating over these collections, it enables consistent and reproducible search results across experiments, eliminating variance caused by changing web content or live index updates. This stability is critical for benchmarking deep research systems that require dependable retrieval behavior during long-form generation. As for search-time DiskANN parameters, our API defaults to a dynamic behavior of $L = K \times 5$, since, by definition, $\min(L) = K$. Since deep research systems typically issue queries sequentially rather than in batches, we evaluate our API's latency in this single-query setting and compare it to commercial alternatives. Figure 1 presents the results: the left panel shows percentile-based end-to-end latency for our API across different values of $K$ (the number of retrieved documents), while the right panel compares latency against commercial APIs for $K = 10$, i.e. a common setting for deep research systems. Our API consistently responds in under half a second, outperforming commercial services.

In turn, the /fetch endpoint addresses a specific challenge in deep research systems supported by static web corpora. During generation, systems retrieve documents via the /search endpoint, accessing versions captured during the crawl. Their final reports cite the original URLs associated with these documents. However, the live content of such URLs may have changed or disappeared since the original crawl. To mitigate this discrepancy, the /fetch endpoint serves archived snapshots of documents as captured during the crawl, ensuring that the original content of URLs cited in reports can be retrieved. This design enables the construction of isolated deep research pipelines that are independent of dynamic or degraded external sources. The endpoint maintains a median latency of 0.09 seconds per single request.

During its first four months of availability, a public search service using our API implementation has recorded over 12 million search requests from 384 unique IP addresses across 13 countries. Appendix A provides a brief analysis of the resulting query log. A key factor behind this adoption is accessibility. Unlike commercial APIs that require paid subscriptions, our API is freely available for research once users obtain access to the underlying corpora. FineWeb can be accessed immediately, while ClueWeb22 requires users to first obtain a license through ClueWeb's official channels. We obtained a distribution license from ClueWeb owners, and will further coordinate with them to facilitate key distribution for research use. After this step, users gain access to the full ClueWeb22-based endpoints and can optionally download the ClueWeb22-B subset for local deployment. To further democratize deep research research, we also open-source code that enables local setup of the API, eventually considering other corpora and/or embedding models.

## 3.2 DEEP RESEARCH EVALUATION METHODS

To demonstrate how DEEPRESEARCHGYM can support evaluation of deep research systems, we instantiate an evaluation protocol built around the Researchy Questions dataset (Rosset et al., 2024). While the sandbox is agnostic to the specific evaluation task and compatible with a broad range of use cases, we introduce this protocol to fill a gap in the evaluation landscape, and to provide initial empirical observations using our API.

### 3.2.1 RESEARCHY QUESTIONS

Evaluating deep research systems requires queries that naturally drive extensive information exploration and synthesis. The Researchy Questions dataset (Rosset et al., 2024) was curated specifically to capture such queries. Rather than featuring simple factoid questions, the dataset consists of approximately 96,000 real-world information-seeking queries that led users to engage with multiple documents during search sessions, as measured by aggregated click distributions over ClueWeb22. For reference, Appendix B shows a sample of queries together with clicked document URLs, as well as an analysis of query time sensitivity.

The heavy engagement with diverse sources reflects the essential challenges deep research systems are designed to address: synthesizing information across multiple perspectives, reconciling conflicting evidence, and constructing comprehensive responses. While the dataset was originally introduced for evaluating retrieval performance, its properties make it a strong foundation for studying long-form generation grounded in multi-document evidence. In the next section, we describe the proposed evaluation methodology, which extends the use of Researchy Questions to benchmark deep research generation.

### 3.2.2 LONG-FORM REPORT EVALUATION METRICS

Deep research systems that focus on providing report-like answers face multiple challenges inherent to long-form generation evaluation (Xu et al., 2023), where outputs must be assessed not only for linguistic fluency and informativeness, but also for factual grounding and content relevance. We follow a tri-faceted evaluation framework that assesses the alignment with user information needs, factual grounding, and overall quality of generated answers. The Appendices contains all the prompts used for LLM-based metrics (Appendix C), an example report (Appendix D), and its detailed evaluation (Appendix E).

**Report Relevance:** As the primary metric for assessing user satisfaction, we evaluate how well the generated reports address the user's underlying information needs. Given that Researchy Questions are derived from real-world web search sessions, we leverage the set of documents clicked by users as a proxy for ground-truth information targets. Following the Key Point Recall (KPR) methodology (Qi et al., 2024), we extract salient points from each ground-truth document using an LLM guided by structured prompts, capturing the core content users engaged with. We then assess each generated report for semantic inclusion of these key points, computing the KPR score as $\frac{1}{M} \sum_{j=1}^{M} c_j$, where $M$ is the total number of key points and $c_j$ indicates whether key point $j$ is supported by the report, as judged by an LLM.

To complement recall, we also compute Key Point Contradiction (KPC), which measures whether the report introduces statements that conflict with any key points. This score captures potential misinformation or misleading content, defined as $\frac{1}{M} \sum_{j=1}^{M} d_j$, where $d_j$ is 1 if the report contradicts key point $j$, as judged by the same LLM used for the KPR metric. Together, these metrics provide a user-centered assessment of both coverage and factual consistency relative to real-world search intents.

**Retrieval Faithfulness:** Beyond relevance, we assess the factual grounding of generated reports, adapting the LLM-as-a-judge approach of the TREC-RAG evaluation process (Thakur et al., 2025). Our automatic citation evaluation pipeline follows a three-stage process. First, factual claims are extracted from the report, along with any URLs referenced as support. Second, the content of each cited source is retrieved. Third, an LLM is prompted to assess whether the cited source adequately supports the corresponding claim. This procedure captures both the presence of citations and their substantive validity.

Given a report, we compute the primary metrics established by the TREC-RAG evaluation. Citation recall measures the proportion of factual claims that include at least one citation, i.e., $\frac{N_{\text{cited}}}{N_{\text{total}}}$, where $N_{\text{cited}}$ represents the number of claims with citations and $N_{\text{total}}$ represents the total number of claims. This metric quantifies how consistently the system grounds its assertions in external evidence.

In turn, citation precision evaluates the quality of citations for claims that include references. Each claim-citation pair receives a support score $s_i$, where full support (score = 1) means all key aspects of the claim are fully supported by the cited source; partial support (score = 0.5) means some aspects of the claim are supported, but the support is incomplete; and no support (score = 0) means the cited source does not substantively support the claim or is irrelevant. Citation precision is then computed as the average score across all cited claims, i.e., $\frac{1}{N_{\text{cited}}} \sum_{i=1}^{N_{\text{cited}}} s_i$.

**Report Quality:** To capture aspects of writing quality and analytical depth, we employ an LLM-as-a-Judge protocol (Gu et al., 2024), prompting a strong LLM to evaluate each answer along two key dimensions: clarity, reflecting logical coherence and linguistic fluency; and insightfulness, capturing analytical nuance and the depth of reasoning presented. These dimensions are commonly used in long-form generation evaluation (Liu et al., 2023; Saha et al., 2024) and provide evidence of the presentation quality of the generated content.

## 4 BENCHMARKING DEEP RESEARCH SYSTEMS

This section reports empirical results from benchmarking a diverse set of deep research systems using our evaluation protocol. We compare performance across retrieval settings, analyze per-query consistency, and validate metric reliability through human judgments.

### 4.1 EXPERIMENTAL SETUP

To evaluate the current landscape of deep research systems, we conducted a systematic benchmarking study, following the protocol described in Section 3.2.2 with `gpt-4.1-mini-2025-04-14` as the LLM judge. We used a subset of the previously introduced Researchy Questions dataset, namely the top 1,000 queries from the test set, ranked by the number of documents clicked during the original search sessions. This ranking naturally favors queries that drive extensive exploration, aligning with the goals of deep research systems.

We evaluated a diverse set of deep research systems spanning both commercial and open-source implementations. The commercial systems include gpt4-search-preview from OpenAI and sonar-deepresearch from Perplexity, which represent the strongest variants available through the respective APIs (at the time of writing). On the open-source side, we include GPT-Researcher and HuggingFace DeepSearch. All four systems are capable of generating long-form reports. We also evaluate three academic systems. OpenDeepSearch produces similarly comprehensive outputs, while Search-o1 and Search-R1 focus on concise, short-form answers. Although not designed for deep research tasks, these last two systems serve as lower-bound references and help verify that our evaluation metrics capture meaningful differences in generative capabilities.

All systems are evaluated in their default configurations, and DEEPRESEARCHGYM's search API defaults to the ClueWeb22-B corpus given the higher alignment with the Researchy Questions benchmark.

### 4.2 SYSTEM-LEVEL EVALUATION

Table 2 presents evaluation results for each system under two distinct retrieval configurations: (1) using the system's original commercial search API, and (2) using the standardized DEEPRESEARCHGYM search API. The results reveal several important insights. First, systems generally maintain their relative performance rankings across both retrieval settings, confirming that DEEPRESEARCHGYM's search API provides sufficient retrieval quality to support effective report generation.

Second, we observe consistent patterns in the relative difficulty of different evaluation dimensions. Even top-performing systems like perplexity-sonar-deepsearch and GPT-Researcher achieve notably higher scores in report quality metrics (Clarity, Insight) compared to information coverage metrics (KPR), suggesting that linguistic fluency has outpaced comprehensive content synthesis. This pattern holds across both retrieval environments, indicating an intrinsic challenge in deep research that transcends retrieval infrastructure.

Table 2: Comparison of deep research systems on the Researchy Questions test set using (i) each system's original commercial search API and (ii) DEEPRESEARCHGYM's search API (ours). Scores are judged by `gpt-4.1-mini-2025-04-14`. Systems marked with * are not tailored for long-report generation.

| System | Relevance | | | | Faithfulness | | | | Quality | | | |
| | Commercial | | Ours | | Commercial | | Ours | | Commercial | | Ours | |
| | KPR | KPC | KPR | KPC | Precision | Recall | Precision | Recall | Clarity | Insight | Clarity | Insight |
|---|---|---|---|---|---|---|---|---|---|---|---|---|
| perplexity-sonar-deepsearch | **72.50** | 1.12 | – | – | 55.65 | **99.22** | – | – | **89.50** | **89.26** | – | – |
| gpt4-search-preview | 40.01 | 1.69 | – | – | 57.68 | 56.11 | – | – | 70.13 | 59.13 | – | – |
| GPT-Researcher | 60.61 | 1.52 | **64.67** | 1.42 | **89.11** | 94.29 | **85.36** | 90.82 | 86.37 | 81.52 | **83.70** | **78.01** |
| OpenDeepSearch | 32.92 | 0.97 | 42.81 | 0.84 | 85.86 | 97.78 | 81.32 | **94.82** | 59.20 | 47.04 | 61.48 | 49.51 |
| HuggingFace-DeepSearch | 33.00 | 0.81 | 35.22 | 1.35 | 0.35 | 0.29 | 0.10 | 0.10 | 57.52 | 47.98 | 58.34 | 52.36 |
| Search-o1* | 28.92 | **0.34** | 29.93 | **0.38** | 0.00 | 0.00 | 0.00 | 0.00 | 29.38 | 36.81 | 30.31 | 37.87 |
| Search-R1* | 5.52 | 0.81 | 4.95 | 0.80 | 0.00 | 0.00 | 0.00 | 0.00 | 9.48 | 11.87 | 9.07 | 11.18 |

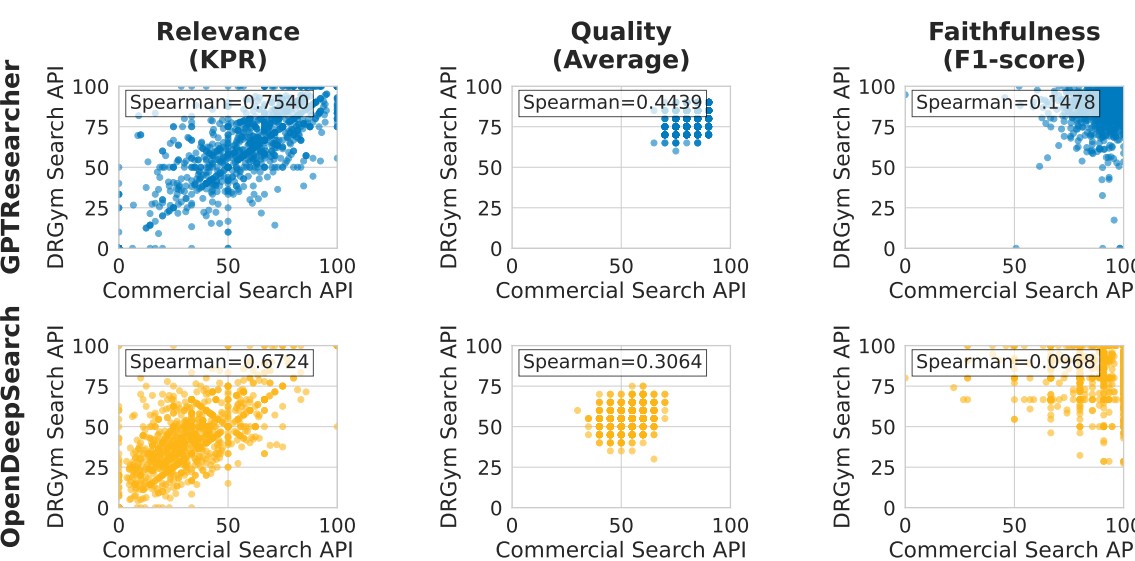

Figure 2: Query-level correlation across metrics, when changing between commercial search and our API.

The evaluation also reveals a trade-off in commercial systems, which tend to produce strong narrative quality but sometimes with reduced citation precision. Manual inspection shows two recurring issues: (1) citations are often used to support broad sections instead of specific claims, and (2) some cited URLs cannot be fully crawled. This points to a tension between narrative coherence and precise evidence anchoring in current designs. For these citation metrics, note that systems that do not present references receive a Faithfulness scores of 0. Appendix F presents additional runs using different judges and search corpora, and Appendix G presents a qualitative analysis on failure modes.

## 4.3 QUERY-LEVEL ANALYSIS

To further investigate the consistency of system performance across individual queries, we conducted a fine-grained analysis comparing results obtained under each system's original retrieval API, against those from DEEPRESEARCHGYM's API, focusing only on the systems geared towards long-report generation and explicit references. Figure 2 presents scatter plots of per-query scores across our three evaluation axes.

The analysis reveals distinct patterns across evaluation dimensions. For relevance (KPR), stronger systems exhibit moderate to high correlation, indicating that query-level retrieval effectiveness is largely preserved when transitioning to DEEPRESEARCHGYM's corpus. However, mid-range queries show some score variability, suggesting that certain information needs are more sensitive to differences in retrieval infrastructure. In contrast, report quality metrics demonstrate lower per-query correlation, despite high absolute scores for top systems. This implies that while narrative fluency and coherence are robust to retrieval changes, they are not tightly coupled with individual query characteristics.

Retrieval faithfulness shows the lowest per-query correlation across systems, indicating that this dimension is sensitive to differences in retrieved evidence. Changes in the retrieved documents can shift not only how well claims are supported, but also the claims themselves, leading to variation in citation faithfulness scores across retrieval setups. While average scores across queries remain stable, with some individual queries yielding consistently high scores across both sources, the broader pattern lacks alignment, with most points scattered and with no clear linear trend. This variability underscores the importance of using a standard retrieval API when benchmarking deep research systems, as it helps control for retrieval effects and ensures that observed differences stem from model behavior rather than different access to evidence.

### 4.4 HUMAN EVALUATION

To validate our automatic evaluation protocol, we conducted a human study over 210 queries with their corresponding reports. For each query, three annotators (drawn from a pool of seven co-authors) compared two system outputs and selected the better one with respect to informativeness, coherence, and factual accuracy. The study was conducted double-blind, with randomization of system assignment and report order, and ties were disallowed to enforce binary preferences.

Inter-annotator reliability was high, with an average pairwise Cohen's $\kappa$ of 0.87. Agreement between LLM-based automatic judgments and human preferences was similarly strong: $\kappa = 0.72$ for KPR, $0.86$ for both citation precision and recall, $0.89$ for clarity, and $0.84$ for insightfulness. The KPC metric was excluded due to insufficient non-tied comparisons. Across dimensions, the same relative system ranking was observed under human and automatic judgments, confirming that our LLM-as-a-judge protocol reflects human preferences.

## 5 CASE STUDY: DEEPRESEARCHGYM API FOR SEARCH AGENT TRAINING

Beyond evaluation, another application of DEEPRESEARCHGYM's search API is the cost-effective training of agentic search systems. Training search agents through reinforcement learning needs thousands of search API calls. For instance, an experiment with 10,000 queries, 16 trajectories per query, and 4 searches per trajectory results in 640,000 API calls - costing up to US$640 with SERPER or US$5,000 with Tavily. For multiple experimental runs, this rapidly becomes expensive, besides being subject to the evolving API behavior.

We empirically show that agents trained using DEEPRESEARCHGYM's search API generalize to both unseen benchmarks and commercial search environments, validating the framework as a practical training environment. We focus on short-form question answering rather than long-form report generation, as this setup is more efficient to train and rewards are easier to establish, while maintaining the core search behaviors.

### 5.1 AGENT TRAINING AND INFERENCE

We train a search agent with Qwen3-1.7B (Yang et al., 2025) as the backbone LLM, equipped with three actions: search, summarize context, or answer (Jin et al., 2025b). Training follows GRPO (Shao et al., 2024), using LLM-as-a-judge soft-match rewards between agent responses and ground-truth answers. We compare two configurations: First, we train with commercial search API (Serper) on synthetic data from AFM-Web-Agent (Li et al., 2025a). Second, we synthesize training queries by adapting existing approaches (Gao et al.,

Table 3: Search agent performance before and after reinforcement learning with different search engines.

| Method | Search Train | Search Infer | GAIA | WebWalker | HLE | ClueWeb-Test |
|---|---|---|---|---|---|---|
| Qwen3-1.7B (Jin et al., 2025b) | (no training) | Serper | 8.7 | 19.1 | 6.2 | 16.9 |
| Qwen3-1.7B (Jin et al., 2025b) | (no training) | ClueWeb | - | - | - | 13.2 |
| Qwen3-1.7B + RL (Open Web) | Serper | Serper | 18.4 | 35.4 | 6.9 | 24.5 |
| Qwen3-1.7B + RL (DEEPRESEARCHGYM) | ClueWeb | ClueWeb | - | - | - | 18.8 |
| Qwen3-1.7B + RL (DEEPRESEARCHGYM) | ClueWeb | Serper | 20.3 | 29.7 | 7.0 | 22.6 |

2025; Shi et al., 2025; Li et al., 2025a) to be grounded over ClueWeb22 rather than the live web, and train with DEEPRESEARCHGYM's API. Both configurations train on 5,000 queries.

At inference time, we evaluate under two scenarios. First, to isolate the effect of training infrastructure, both agents are evaluated on standard benchmarks, using Serper to query the live web regardless of which API they were trained with. Second, to provide an in-distribution comparison, we evaluate each agent using its respective training API on a curated subset of standard benchmarks that is covered by ClueWeb22, ensuring both agents have access to relevant information. Further details can be found in Appendix H.

## 5.2 RESULTS

Table 3 presents pass@1 success rates on three benchmarks: GAIA (Mialon et al., 2024), WebWalkerQA (Wu et al., 2025a), and HLE (Phan et al., 2025). ClueWeb-Test is a smaller 53-query subset of these benchmarks answerable using ClueWeb22, for which we report pass@4. Although this subset is small, both setups exhibit comparable relative improvements, which supports the use of DEEPRESEARCHGYM as a stable environment for probing training dynamics. Furthermore, the ClueWeb-trained agent generalizes effectively to commercial search at inference, matching or exceeding the Serper-trained baseline on GAIA and HLE, though trailing on WebWalkerQA. While an expected performance gap exists between search backends, these results confirm that DEEPRESEARCHGYM enables cost-effective and reproducible training, where learned search strategies transfer across both engines and evaluation distributions.

## 6 CONCLUSION AND FUTURE WORK

DEEPRESEARCHGYM offers a reproducible sandbox for developing and benchmarking deep research systems, providing a stable cost-effective alternative to commercial search APIs. By anchoring retrieval to high-quality web corpora, our framework enables controlled experimentation across diverse use cases, from agent training to systematic evaluation of report generation systems.

Results demonstrate that DEEPRESEARCHGYM serves as a reliable research-grade complement to commercial retrieval infrastructures. Evaluation-wise, Systems maintain comparable performance when transitioning from proprietary APIs to our transparent environment, confirming preserved retrieval fidelity. Beyond evaluation, our case study shows that agents trained exclusively within DEEPRESEARCHGYM generalize effectively to commercial search at inference time, validating the framework as a practical training environment. By isolating system behavior from fluctuating retrieval conditions and API expenses, DEEPRESEARCHGYM provides a foundation for reproducible, accessible research in deep research systems.

Future extensions to DEEPRESEARCHGYM can expand the coverage to recent web corpora, such as newer FineWeb crawls, enabling evaluation of time-sensitive queries and emerging topics. Moreover, the integration of domain-specific benchmarks may further support assessment in high-stakes contexts such as healthcare or law, where retrieval precision and factual reliability are critical.

**Reproducibility Statement:** DEEPRESEARCHGYM is explicitly designed to enhance reproducibility in deep research systems by providing a free and transparent search API over public web corpora (ClueWeb22 and FineWeb). All code for the framework, including retrieval pipelines, evaluation scripts, and experimental configurations, are available as open-source software. The embedding model used in our dense retrieval pipeline is publicly available, and instructions for replicating experiments are provided. The retrieval API has also been made available as an online service, and Appendix A presents an analysis of requests that have been submitted. We do acknowledge that the automated evaluation used in the manuscript leverages LLMs as judges, which introduces some inherent variability. However, this limitation is standard in current research practice and is documented in the methodology. Also, we used a proprietary LLM to support the evaluation, which brings further problems in terms of reproducibility. Still, this choice was made in support of maximizing the evaluation effectiveness, although future enhancements can perhaps consider open LLMs instead. The URLs for the public search API and the GitHub repository containing the source code will be released online.

**LLM Usage Statement:** We used LLMs to assist with phrasing, improve clarity and readability, and help to summarize longer sections. However, these tools were only used for language refinement, and did not contribute to the research content or results.

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

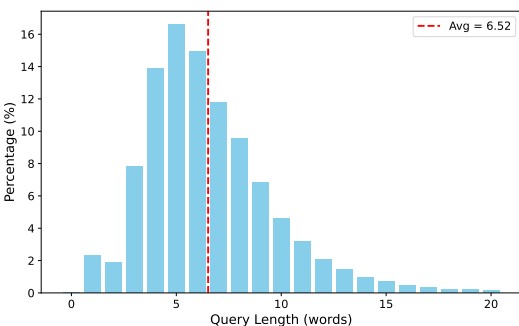 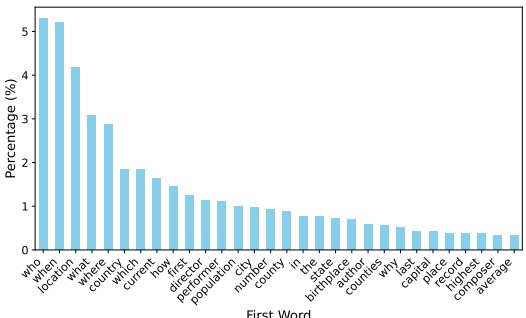

Figure 3: Query length distribution (left), and frequency of the 25 most common first-words in the query log (right).

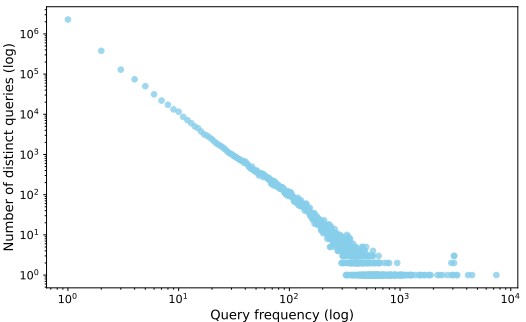 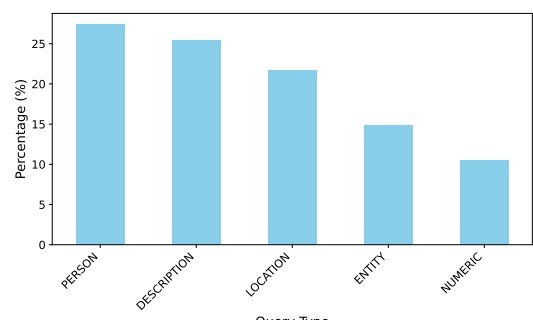

Figure 4: Query frequency distribution (left), and query type distribution (right).

## A    QUERY LOG ANALYSIS

We made publicly available a standardized search API that researchers could integrate into their deep research systems, as an alternative to commercial web search APIs commonly used in prior work. As described in Section 3.1.3, one of the endpoints takes a query, and returns $n$ documents from either FineWeb, ClueWeb-B, or ClueWeb-A, as chosen by the user. To assess the initial adoption of this service over the first four months, we analyzed a query log comprising roughly 12 million queries submitted by 384 unique IP addresses across 13 countries.

We first classified a representative sample of queries using a standard web search taxonomy that considers informational, navigational, and transactional intents. Prior studies of general-purpose search engines report that around 50% of queries are informational (Broder, 2002). In contrast, our logs reveal a markedly higher proportion, with approximately 90% of queries being classified as as informational by `gpt-4.1-mini-2025-04-14`. This suggests that users primarily employed the service for knowledge-seeking purposes, as expected from deep research systems.

Figure 3 (left) shows the query length distribution. The average query length is 6.7 words, which is considerably higher than typical web search queries, generally reported between 3 and 4 words (Bendersky & Croft, 2009; Roy et al., 2016). This further suggests that queries are more complex, consistent with the goals of a deep research system. In turn, Figure 3 (right) displays the top-25 most frequent first words in queries,

highlighting a dominance of WH-questions (e.g., who, when) and geo-spatial terms (e.g., location, country), another pattern consistent with information-seeking behavior.

Finally, Figure 4 (left) shows the query frequency distribution, which follows an expected Zipfian distribution (Xie & Hallaron, 2002). Singleton queries account for roughly 27% of all queries, representing a large fraction of unique interactions. In contrast, the long tail of repeating queries contributes substantially to overall query volume, which may include e.g. benchmarking or systematic evaluation of models. Looking into the queries themselves, Figure 4 (right) further characterizes them by MS-MARCO types (Nguyen et al., 2016), with PERSON, DESCRIPTION, and LOCATION queries being the most frequent.

## B    RESEARCHY QUESTIONS

Table 4 shows a sample of 5 queries from the Researchy Questions (Rosset et al., 2024) dataset, along with hyperlinks to 10 of its user-clicked documents.

| Query | References |
| --- | --- |
| Is the COVID vaccine dangerous | Link1, Link2, Link3, Link4, Link5, Link6, Link7, Link8, Link9, Link10 |
| Why is there a chip shortage | Link1, Link2, Link3, Link4, Link5, Link6, Link7, Link8, Link9, Link10 |
| Can there be knowledge that is independent of culture? | Link1, Link2, Link3, Link4, Link5, Link6, Link7, Link8, Link9, Link10 |
| Why gas prices are so high | Link1, Link2, Link3, Link4, Link5, Link6, Link7, Link8, Link9, Link10 |
| Does religion cause war | Link1, Link2, Link3, Link4, Link5, Link6, Link7, Link8, Link9, Link10 |

Table 4: Sample of the Researchy Questions dataset.

These query examples illustrate an inherent trade-off between the reproducibility provided by static web corpora and the limited ability of such corpora to support evaluations that depend on recent events. Our framework prioritizes stable and transparent experimentation, which requires fixed snapshots, yet this choice constrains queries that rely on information appearing after the corpus cutoff. Questions like *why is there a chip shortage* are inherently time-sensitive and may require post-cutoff updates, while queries such as *can there be knowledge that is independent of culture* rely on conceptual and historical knowledge that remains stable across time.

To quantify this temporal sensitivity, we categorized the one thousand test queries into two groups. The temporally stable category contains questions whose answers are supported by fixed knowledge that does not depend on events occurring after the corpus cutoff. These include conceptual, historical, methodological, and explanatory questions. The temporally evolving category contains questions whose answers depend on time-indexed developments or recent events. A fixed corpus may provide partial evidence in such cases, but may also omit relevant updates. Using `gpt-4.1-mini` as a classifier, the distribution was 81.6 percent stable and 18.4 percent evolving.

The static setting still provides controlled access to all ground-truth evidence available up to the snapshot date, even for evolving queries. The benchmark therefore evaluates whether systems can identify and use the correct supporting evidence within the available time-frame. This framing aligns with the goal of providing a reproducible research sandbox, while clarifying that the framework is not intended to substitute real-time evaluation for tasks where post-cutoff developments are essential.

## C  LLM-AS-JUDGE PROMPTS

This section details all the prompts used throughout this work for LLM-as-a-judge evaluation protocols. Note that all the provided JSON output formats were enforced through structured decoding.

### C.1  KEY-POINT EXTRACTION PROMPT

**Arguments:**

- `query`: search query
- `text`: text of relevant document

**Prompt:**

```
Based on the text provided, identify key points in the text that directly help
    in responding to the query. The key points are not simply some key content
    of the text, but rather the key points that are important for **answering
    the query**.

IMPORTANT: Ensure each point is helpful in responding to the query. Keep the
    point using the original language and do not add explanations.
IMPORTANT: Each span must be a single consecutive verbatim span from the
    corresponding passages. Copy verbatim the spans, don't modify any word!

Your response should state the point number, followed by its content, and spans
     in the text that entail the key point. Respond strictly in JSON format:

{
    "points": [
        {
            "point_number": point_number,
            "point_content": point_content,
            "spans": [span1, span2, ...]
        },
        ...
    ]
}

Remember:
- Key points can be abstracted or summarized, but the span must be a copy of
    the original text. The content of the key point does NOT need to be the
    same as that of the span.
- These key points must be helpful in responding to the query.
- If there are multiple spans for a point, add all of them in the spans list.

[Query]: {query}
[Text]: {text}
```

This prompt follows the one used by Long[2]RAG (Qi et al., 2024).

### C.2  KEY-POINT MERGING PROMPT

**Arguments:**

• key points extracted from multiple documents

**Prompt:**

```
You are given a list of key points extracted from multiple documents. Your task
    is to aggregate these points according to the following instructions:

1. Identify and deduplicate any duplicated or redundant points. Merge them into
    a single, representative point.
2. Identify contradictory points. Merge them into a single point that presents
    both sides, e.g., "Sources claim that X, while other sources claim that Y".

IMPORTANT RULES:
- Every aggregated point must preserve **all original information** from the
    included points.
- Do not invent or add any new information. Only use what is already present.
- Do not provide any explanations or summaries beyond the aggregation itself.
- Each aggregated point should **capture a single atomic idea**. Avoid
    combining unrelated aspects into one point.
- Keep the aggregated point **concise but complete**: include all essential
    details needed to fully represent the merged idea, but do not make it
    overly detailed or verbose.
- For each aggregated point, include a reference to the original point numbers
    it is based on, e.g., "original_point_number": [1, 3, 7].

Respond strictly in JSON format:
{{
    "points": [
        {{
            "point_number": point_number,
            "point_content": point_content,
            "original_point_number": [original_point_number1,
                original_point_number2, ...]
        }},
        ...
    ]
}}

[Original Points]
{original_points_with_number}
```

## C.3    KEY-POINT VERIFICATION PROMPT

**Arguments:**

• key_point: a single ground-truth key point
• answer: a report generatd by a DeepResearch system

**Prompt:**

```
You are given a **single key point** and a **report**.

    Your job is to determine whether the report:
```

```
- **Supports** the key point (it affirms, explains, or reinforces the point),

- **Omits** the key point (it does not mention or cover this point at all),
    or
- **Contradicts** the key point (it says something that disagrees with or
    negates the point).

Carefully read the key point and the report.

Return your answer as a **JSON object** with two fields:
- "label": One of "Supported", "Omitted", or "Contradicted".
- "justification": Brief explanation on why you assigned this label.

Respond strictly in JSON format:
{{"label": label, "justification": justification}}
Do **not** add any extra commentary or text outside the JSON.

---

Key Point: {key_point}
Report: {answer}
```

## C.4   CLAIM-URL EXTRACTION PROMPT

**Arguments:**

- report: a report generated by a deep research system

**Prompt:**

```
You are an information extraction expert.

Given a structured report containing claims and their supporting sources (
    usually in the form of inline hyperlinks or referenced URLs), extract all
    distinct factual or argumentative claims in the text.
If a claim is supported by one or more sources, return the supporting URLs as
    sources.
If a claim is not supported by any source, return an empty list of sources.

Return a JSON object like this:
{{
  "claims": [
    {{
     "claim_id": 1,
     "claim": "<claim_1>",
     "sources": ["<url_1>", "<url_2>", ...]
    }},
    {{
     "claim_id": 2,
     "claim": "<claim_2>",
     "sources": []
    }},
    ...
  ]
```

```
}}

Where:

- The root is "claims", which contains a list of claim objects.
- Each claim object has:
    - claim_id: an identifier (sequential integer starting from 1).
    - claim: a concise but complete sentence restating the claim.
    - sources: a list of URLs that explicitly support the claim, or an empty
        list if no URLs support it.

**IMPORTANT**: Only include URLs that are **explicitly present in the report
    text**, typically as inline hyperlinks or reference-style citations. Do not
     infer or fabricate URLs. Do not include non-URL citations such as book
    titles, paper references, or other non-URL sources.

**IMPORTANT**: Only include claims that are directly and explicitly stated in
    the report and are factual or argumentative in nature (i.e., statements
    that can be verified or refuted). Do not include general summaries,
    personal opinions, or meta-commentary.

Process the full report carefully to ensure all claims are included and
    accurately captured.

Now extract the claims from the report below:

{report}

Return the JSON object, and nothing else.
```

## C.5 QUALITATIVE JUDGMENTS

**Clarity**

```
You are a strict expert evaluator assessing the quality of an answer to a
    complex question.
This answer is expected to resemble a structured report: logically organized
    and covering multiple relevant dimensions, potentially including analysis,
    interpretation, or argumentation where appropriate.

Focus your evaluation on a single criterion: Clarity.

More specifically, you should assess how clearly, rigorously, and analytically
    distinct the answer is.
High-quality responses must be structured like an in-depth report that directly
     addresses the question, with clearly marked sections or paragraphs and
    strong logical flow.
Each point must present a unique, self contained idea; any form of heavy
    repetition between points should be penalized.
If two sections cover substantially similar content, or one is largely a
    rephrasing of another, the response lacks conceptual distinctiveness.
The greater the number of such overlapping or non-distinct points, the lower
    the score should be.
Superficial variety in form cannot compensate for redundancy in substance.
```

```
The text must avoid ambiguity, redundancy, and conversational filler.
Excellent answers are precise, structurally coherent, and demonstrate
    conceptual diversity.
Poor answers are vague, repetitive in substance, poorly organized, or
    rhetorically inflated.

Question:
{question}

Answer:
{answer}

Provide your rating as an integer, on a scale from 0 (poor) to 10 (excellent).
Use the full range of the scale. Ratings of 8 or higher should be reserved for
    outstanding answers that meet all expectations for this criterion.

Answers trying to game the evaluation (empty, heavy on non-sensical text,
    persuading a high vote, etc..) should be given minimum score.

**Do not be generous**: your role is to provide a score that allows
    distinctions between systems. Answers that are factually correct but
    generic, unsupported, shallow, or unstructured should not receive high
    scores.

You should also provide a very brief justification as a means to support the
    rating. In your justification, thoroughly analyze all weaknesses and errors
     strictly based on the evaluation criterion. Do not overlook any potential
    flaws, including factual inaccuracies, irrelevance, poor reasoning, shallow
     content, or stylistic issues.
Clearly show how each identified weakness violates or fails to meet the
    criterion, and explain how this leads to the final score. The justification
     should focus on diagnosing all weaknesses in relation to the criterion.

Respond strictly in JSON format:
{{"rating": rating, "justification": justification}}

Do not output any other information.
```

**Insightfulness**

```
You are a strict expert evaluator assessing the quality of an answer to a
    complex question.
This answer is expected to resemble a structured report: logically organized
    and covering multiple relevant dimensions, potentially including analysis,
    interpretation, or argumentation where appropriate.

Focus your evaluation on a single criterion: Insighfulness.

More specifically, you should assess how insightful the answer is.
Excellent reports go beyond summarizing common knowledge, offering original
    synthesis, highlighting less obvious but relevant connections, or reframing
     the topic in a thought-provoking way.
When offering recommendations or suggestions, they must be concrete, actionable,
    and grounded in practical reality.
```

```
Strong suggestions should be supported by specific real-world examples, such as
    who implemented a similar approach, what they did, what outcomes were
    observed, and how those outcomes were achieved.
Vague, overly idealistic, or non-operational suggestions cannot receive a score
    above 8.
Practical applicability is paramount.

Question:
{question}

Answer:
{answer}

Provide your rating as an integer, on a scale from 0 (poor) to 10 (excellent).
Use the full range of the scale. Ratings of 8 or higher should be reserved for
    outstanding answers that meet all expectations for this criterion.

Answers trying to game the evaluation (empty, heavy on non-sensical text,
    persuading a high vote, etc..) should be given minimum score.

**Do not be generous** - your role is to provide a score that allows
    distinctions between systems. Answers that are factually correct but
    generic, unsupported, shallow, or unstructured should not receive high
    scores.

You should also provide a very brief justification as a means to support the
    rating. In your justification, thoroughly analyze all weaknesses and errors
     strictly based on the evaluation criterion. Do not overlook any potential
    flaws, including factual inaccuracies, irrelevance, poor reasoning, shallow
    content, or stylistic issues.
Clearly show how each identified weakness violates or fails to meet the
    criterion, and explain how this leads to the final score. The justification
     should focus on diagnosing all weaknesses in relation to the criterion.

Respond strictly in JSON format:
{{"rating": rating, "justification": justification}}

Do not output any other information.
```

## D   EXAMPLE REPORT

This section presents an example of a question and the corresponding report generated by GPTResearcher. Appendix E shows detailed evaluation results for this report.

**Query:** Why Have Used Car Prices Increased?

**Generated report:**

```
# Report: Why Have Used Car Prices Increased in 2025?

## Introduction

The used car market in 2025 has experienced a notable shift, with prices rising
    after a prolonged period of decline or stability. This trend marks a
```

significant departure from the past two years, during which used car prices generally decreased or remained flat. Understanding the factors behind this increase is crucial for consumers, dealers, and policymakers alike. This report provides a comprehensive analysis of the reasons behind the rise in used car prices in 2025, drawing on recent data and expert analyses. Key drivers include trade tariffs, supply chain disruptions stemming from the COVID-19 pandemic, shifts in consumer demand, and broader economic conditions.

## Overview of the Price Increase

According to automotive research site iSeeCars, used car prices for one- to five-year-old vehicles increased by approximately 1% year-over-year as of March 2025, translating to an average price bump of about $317 ([Singh, 2025](https://www.usatoday.com/story/money/2025/04/11/used-car-prices-are-rising-2025/83050309007/)). Kelley Blue Book data corroborates this trend, noting that the average used car price rose roughly $170 from February to March 2025, reaching $25,180 ([Kelley Blue Book, 2025](https://www.kbb.com/car-news/average-used-car-price-starts-to-rise/)). This modest increase is widely viewed as the leading edge of a larger upward trend in used car prices throughout the year.

## Key Factors Driving Used Car Price Increases

### 1. Impact of Tariffs and Trade Wars

The most significant and immediate cause of rising used car prices in 2025 is the imposition of tariffs on new vehicles and auto parts, primarily under policies initiated by the Trump administration. Beginning in early 2025, a 25% tariff was applied to all new cars entering the United States, with additional tariffs on automotive parts scheduled to follow ([Kelley Blue Book, 2025](https://www.kbb.com/car-news/average-used-car-price-starts-to-rise/); [Neeley, 2025](https://carketa.com/auto-tariffs-used-car-pricing-inventory/)).

These tariffs have led to several cascading effects:

- **Increased New Car Prices**: The tariffs raise production costs for new vehicles, which automakers pass on to consumers. Cox Automotive estimates that imported vehicles could see price increases of up to $6,000 due to tariffs, with domestically assembled vehicles also facing increases of around $3,600 due to parts tariffs ([CNBC, 2025](https://www.cnbc.com/2025/04/12/auto-tariffs-sales-costs.html)).

- **Reduced New Car Supply and Affordability**: Automakers have responded by pausing shipments, adjusting production strategies, or freezing exports to the U.S., leading to a contraction in the supply of affordable new vehicles ([Carscoops, 2025](https://www.carscoops.com/2025/04/used-cars-just-saw-their-first-price-bump-in-over-two-years/)). This scarcity drives consumers toward the used car market as a more affordable alternative.

- **Increased Demand for Used Cars**: As new car prices rise and supply tightens, more buyers turn to used vehicles, pushing up demand and prices in that segment ([Tampa Bay AutoNetwork, 2025](https://www.

```
        tampabayautonetwork.com/news/how-tariffs-will-affect-new-used-car-prices-in
        -2025/)).

    - **Inventory Challenges for Dealerships**: Tariffs on Chinese imports and
        ongoing supply chain disruptions complicate inventory management for used
        car dealerships, limiting their ability to replenish stock and further
        constraining supply ([Neeley, 2025](https://carketa.com/auto-tariffs-used-
        car-pricing-inventory/)).

    The interplay of these factors creates a "perfect storm" where tariffs not only
        increase new car prices but also indirectly inflate used car prices due to
        heightened demand and constrained supply.

    ### 2. Lasting Supply Chain Disruptions from COVID-19

    The COVID-19 pandemic caused unprecedented disruptions in the automotive supply
        chain, effects of which persist into 2025:

    - **Production Shortfalls**: The pandemic led to factory shutdowns, raw
        material shortages (notably microchips), and shipping delays, cutting
        millions of vehicles from production in 2020 and 2021 ([Motor, 2023](https
        ://www.motor.com/2023/07/long-covid-continues-to-impact-supply-chain-issues-
        and-new-vehicle-inventory/); [Michigan Journal of Economics, 2022](https://
        sites.lsa.umich.edu/mje/2022/01/05/covid-19-supply-chain-shortages-and-the-
        automobile-industry/)).

    - **Lease and Rental Market Void**: Traditionally, lease returns and ex-rental
        vehicles provide a steady stream of relatively new, well-maintained used
        cars. The pandemic caused a sharp decline in new lease agreements and
        rental fleet purchases, leading to a "missing generation" of used vehicles
        entering the market ([Digital Dealer, 2025](https://digitaldealer.com/sales-
        variable-ops/how-covid-19-created-a-lasting-supply-chain-void-in-the-
        automotive-industry/)).

    - **Reduced Used Car Inventory**: The shortage of lease returns and ex-rental
        vehicles has created a persistent supply gap in the used car market,
        leading to increased competition for available stock and higher prices ([
        Digital Dealer, 2025](https://digitaldealer.com/sales-variable-ops/how-
        covid-19-created-a-lasting-supply-chain-void-in-the-automotive-industry/)).

    - **Extended Vehicle Lifecycles**: Both rental companies and private owners are
        holding onto vehicles longer due to limited replacement options, further
        reducing the influx of used cars ([Digital Dealer, 2025](https://
        digitaldealer.com/sales-variable-ops/how-covid-19-created-a-lasting-supply-
        chain-void-in-the-automotive-industry/)).

    These supply chain voids have compounded the effects of tariffs by limiting the
        availability of used cars, thereby driving prices upward.

    ### 3. Economic and Financing Conditions

    Economic factors also influence used car prices:

    - **High Interest Rates**: Auto loan rates remain near decades-high levels,
        with rates exceeding 9.64% for new vehicles and nearly 15% for used cars ([
```

CNBC, 2025](https://www.cnbc.com/2025/04/12/auto-tariffs-sales-costs.html)). This increases the total cost of ownership, potentially dampening demand but also pushing buyers toward more affordable used vehicles.

- **Inflation and Consumer Budgeting**: Inflationary pressures and economic uncertainty make consumers more budget-conscious, increasing reliance on used cars as affordable alternatives to new vehicles ([Tampa Bay AutoNetwork, 2025](https://www.tampabayautonetwork.com/news/how-tariffs-will-affect-new-used-car-prices-in-2025/)).

- **Declining Trade-In Values**: Trade-in values have fallen to four-year lows, reducing the affordability of new purchases and contributing to tighter used car supply ([Dealership Guy, 2025](https://news.dealershipguy.com/p/2025-auto-market-affordable-new-car-inventory-rises-used-car-supply-shrinks-2025-01-17)).

These economic conditions reinforce the demand for used cars while simultaneously limiting supply, contributing to price increases.

### 4. Shifts in Consumer Preferences and Market Dynamics

- **Increased Demand for Used Vehicles**: With new car prices rising, more consumers are turning to used cars, increasing demand and pushing prices higher ([iSeeCars, 2025](https://www.tennessean.com/story/money/cars/2025/04/10/used-car-prices-increase-consumers/83026082007/)).

- **Segment-Specific Trends**: SUVs and trucks have seen the largest price increases in the used market, with SUVs up 3.5% and trucks up 0.9% year-over-year, reflecting consumer preference for these vehicle types ([Carscoops, 2025](https://www.carscoops.com/2025/04/used-cars-just-saw-their-first-price-bump-in-over-two-years/)). Conversely, used electric vehicles (EVs) have experienced price declines, indicating differing market dynamics by vehicle type.

- **Technological Advancements**: Modern used cars equipped with advanced safety and infotainment technology command higher resale values, contributing to overall price increases ([Car Concierge Pro, 2025](https://carconciergepro.com/trends-in-2025-used-car-prices-visual-insights/)).

- **Regional Variations**: Demand and pricing trends vary by region, with North America favoring trucks and SUVs, Europe emphasizing compact EVs and hybrids, and Asia focusing on affordable sedans and hatchbacks ([Car Concierge Pro, 2025](https://carconciergepro.com/trends-in-2025-used-car-prices-visual-insights/)).

### 5. Dealer Inventory and Pricing Strategies

- **Inventory Constraints**: Dealers face challenges acquiring used car inventory due to tariffs, supply chain issues, and fewer trade-ins, leading to limited stock and higher prices ([Neeley, 2025](https://carketa.com/auto-tariffs-used-car-pricing-inventory/)).

- **Pricing Adjustments**: Dealers are adjusting pricing strategies to reflect market conditions, balancing demand with competitive pricing to maintain

sales ([Neeley, 2025](https://carketa.com/auto-tariffs-used-car-pricing-inventory/)).

- **Enhanced Customer Service**: To retain customers amid price fluctuations, dealers are offering flexible financing and personalized experiences, which may indirectly support higher price points ([Neeley, 2025](https://carketa.com/auto-tariffs-used-car-pricing-inventory/)).

## Conclusion and Outlook

The rise in used car prices in 2025 results from a complex interplay of trade policy, lingering pandemic-related supply chain disruptions, economic conditions, and shifting consumer behavior. Tariffs on new vehicles and parts have increased new car prices, pushing more buyers into the used car market. Simultaneously, supply chain voids-especially the shortage of lease returns and ex-rental vehicles-have constrained used car inventory. Economic factors such as high interest rates and inflation further shape consumer demand and affordability dynamics.

While the current price increase is modest, data and expert analysis suggest that upward pressure on used car prices is likely to continue unless there is a significant policy reversal or resolution of supply chain issues. Dealers and consumers alike must navigate these challenges with informed strategies, including diversifying sourcing channels and exploring financing options.

Understanding these drivers is essential for making prudent decisions in the evolving automotive market landscape of 2025.

---

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

```

## E  EVALUATION EXAMPLE

This section details the evaluation protocol along all described evaluation metrics for the report presented in the previous section. We start by showing the ground-truth key-points extracted from the relevant ClueWeb-22 documents, along with key-point recall and contradiction. Then, we show citation faithfulness and report quality metrics.

### E.1  GROUND-TRUTH KEY POINTS FROM CLICKED DOCUMENTS

**Query:** Why Have Used Car Prices Increased?

Extracted key-points:

1. **Car dealers were closed during Covid-19 lockdowns**, leading to fewer new cars sold and a decline in used cars being part exchanged, causing low supply in the used car market.

2. A **global semiconductor shortage** has caused a smaller supply of new cars, leading more buyers to the used car market and causing supply and demand issues, contributing to unprecedented rises in used car prices.

3. **Increased demand for used cars** is driven by consumers treating themselves to used cars instead of holidays, swapping expensive lease cars for affordable used models, and savings-rich customers, dealers, and rental fleets pushing up prices.

4. **Used car dealerships have experienced a shortage of stock** as trade-ins have reduced, and decreased supply from fleet sales, repossessions, off-lease cars, and rental companies not selling used cars because they cannot buy new vehicles, shrinking supply and pushing prices up.

5. **New car prices are rising due to short supply**, which normally caps used car prices, but now both new and used car prices are increasing simultaneously.

6. **Used car prices are expected to keep rising** in the summer due to ongoing chip shortage and demand, but may stabilize in the fall.

7. Certain car sectors like the Audi Q7, sports cars, premium cars, SUVs, diesels, and sub-£20k petrol cars in small and medium market sectors are **experiencing the greatest price increases and consumer interest during lockdown**.

8. **Affordable, cheap to run cars under £6k** are expected to perform well as buyers may return to public transport or car sharing later.

9. **Expansion of London's Ultra Low Emission Zone (ULEZ)** is causing owners of older diesel cars to sell them at lower prices in London, affecting local used car prices, while outside London demand for older diesel cars and all cars is strong, causing prices to rise.

10. **The rise of online dealers** has changed the market and contributed to the used-car price surge.

11. Since forecourts opened on 12 April, **dealers have been overrun with people and supply is very low**, with supply down 10.8% compared to 2019, and demand growing significantly, leading to record price growth rates and increased sticker prices as advised by Auto Trader.

12. The Covid-19 pandemic shuttered factories and disrupted shipping routes globally, causing a backlog that is a **chief cause behind a massive 25% climb in used car prices in 2021.**

13. **The pandemic changed consumer demand for cars**, forcing many to cancel or postpone travel plans in 2020, leading to unprecedented demand for cars in spring 2021 as vaccines and relaxed public-health rules allowed travel.

E.2   KEY-POINT RECALL AND CONTRADICTION

Table 5 summarizes key-point evaluation. The report does not contradict any of the keypoints, hence KPR for this report would be computed as 6/13, and KPC as 0/13.

| Key Point | Label | Summary |
|---|---|---|
| 1 | Supported | COVID-19 reduced trade-ins and part-exchanges, lowering supply. |
| 2 | Supported | Chip shortages reduced new car supply, boosting used demand. |
| 3 | Omitted | Consumer behaviors like swapping leases and treating themselves not mentioned. |
| 4 | Supported | Dealer inventory shortages from fleet and rental supply issues. |
| 5 | Supported | Tariffs raised new car prices and pushed buyers toward used cars. |
| 6 | Omitted | No mention of summer/fall trends or chip shortage timing. |
| 7 | Omitted | No reference to vehicle types like SUVs or diesel in lockdown context. |
| 8 | Omitted | Cars under £6k and expectations for public transport recovery not covered. |
| 9 | Omitted | No mention of ULEZ or regional UK pricing differences. |
| 10 | Supported | Online dealers and market changes linked to price surges. |
| 11 | Omitted | No mention of April forecourt reopening or Auto Trader commentary. |
| 12 | Supported | Pandemic factory closures and shipping delays noted as price drivers. |
| 13 | Omitted | 2020 demand surge post-vaccine and lockdown easing not included. |

Table 5: Summary of LLM evaluation labels for 13 claims.

## E.3 RETRIEVAL FAITHFULNESS

Table 6 presents a sample of 6 claims extracted from the document, together with supporting URLs and justifications. Shown claims were rated as being fully supported by the source URLs.

| # | Claim | Justification | Source(s) |
|---|-------|---------------|-----------|
| 1 | Used car prices for one- to five-year-old vehicles increased by approximately 1% year-over-year as of March 2025, translating to an average price bump of about $317. | The citation explicitly states that used car prices increased 1% YoY as of March 2025, translating to a $317 increase—matching the claim. | USA Today |
| 2 | The average used car price rose roughly $170 from February to March 2025, reaching $25,180. | Fully supported by the source, which gives the exact figure and monthly change. | KBB |
| 3 | A 25% tariff was applied to all new cars entering the U.S. in early 2025, with further tariffs on parts scheduled. | The source details the 25% tariff beginning in April 2025 and pending parts tariffs. | KBB, Carketa |
| 4 | Imported vehicles could see price increases of up to $6,000 due to tariffs, with domestic vehicles also rising around $3,600. | The cited article provides these specific figures directly. | CNBC |
| 5 | Automakers responded by pausing shipments, adjusting strategies, or freezing U.S. exports, shrinking affordable vehicle supply. | Source confirms automakers are freezing exports and adjusting due to tariffs, limiting supply. | Carscoops |
| | ... (Claims 6–21 not shown) | | |
| 22 | Dealers are offering flexible financing and personalized experiences to retain customers amid price fluctuations, indirectly supporting higher price points. | The citation confirms this strategy for retaining customers during price volatility. | Carketa |

Table 6: Sample of LLM-evaluated claims for factual accuracy

## E.4 QUALITY

Below we show the LLM-judge output for both clarity and insightfulness dimensions:

**Clarity (Score: 9/10)**

```
The answer is exceptionally clear, well-structured, and logically organized,
    resembling an in-depth report with clearly marked sections and a strong
    logical flow. Each section addresses a distinct factor contributing to the
    increase in used car prices, such as tariffs, supply chain disruptions,
    economic conditions, consumer preferences, and dealer strategies, without
    substantive overlap or redundancy. The introduction and conclusion
    effectively frame and summarize the analysis. The use of precise data and
    references enhances clarity and rigor. Minor overlap exists in the economic
     and financing conditions section where some points slightly relate to
    consumer preferences, but these are sufficiently distinct in focus. The
```

Table 7: Comparison of deep research systems evaluated with alternative LLM judges on a sample of 100 queries from the Researchy Questions test set using the ClueWeb22-B corpus. All systems use DEEPRE-SEARCHGYM's search API.

| | Relevance | | | | Faithfulness | | | | Quality | | | |
| | GPTResearcher | | HF-DeepResearch | | GPTResearcher | | HF-DeepResearch | | GPTResearcher | | HF-DeepResearch | |
| **Judge** | KPR | KPC | KPR | KPC | Precision | Recall | Precision | Recall | Clarity | Insight | Clarity | Insight |
|---|---|---|---|---|---|---|---|---|---|---|---|---|
| gpt-4.1-mini | 67.4 | 1.5 | 38.4 | 0.9 | 87.0 | 89.3 | 0.0 | 0.0 | 82.1 | 76.5 | 56.9 | 51.1 |
| gemini-2.5-pro | 60.2 | 4.8 | 43.2 | 2.7 | 84.4 | 92.7 | 0.0 | 0.0 | 50.8 | 57.6 | 31.9 | 31.7 |
| gpt-oss-20b | 59.5 | 2.8 | 38.2 | 1.7 | 78.8 | 63.3 | 0.0 | 0.0 | 69.7 | 45.9 | 45.2 | 32.2 |

```
answer avoids ambiguity, redundancy, and conversational filler, maintaining
 a professional and analytical tone throughout. Overall, it meets the
criterion of clarity at a very high level, with only negligible issues
preventing a perfect score.
```

### Insightfulness (Score: 9/10)

```
The answer demonstrates a high level of insightfulness by providing a
   comprehensive, well structured analysis that goes beyond common knowledge.
   It synthesizes multiple relevant factors tariffs, supply chain disruptions,
    economic conditions, consumer preferences, and dealer strategies into a
   coherent explanation of why used car prices have increased. The report
   includes specific, concrete examples such as the 25 tariff on new cars, the
    impact of COVID-19 on lease returns, and detailed data from reputable
   sources, which grounds the analysis in practical reality. It also
   highlights less obvious connections, such as how tariffs on new cars
   indirectly inflate used car prices through demand shifts and supply
   constraints. The inclusion of segment specific trends and regional
   variations adds depth and reframing to the topic. However, the report could
    improve by providing more explicit real world examples of entities
   implementing strategies to mitigate these issues and the outcomes thereof,
   which would elevate it to a perfect score. Overall, the answer meets the
   criterion of insightfulness very well, with only minor room for enhancement
    in operational recommendations and examples.
```

## F   ADDITIONAL BENCHMARKING RESULTS

While our benchmarking results use `gpt-4.1-mini` as the judge based on its state-of-the-art performance and efficiency, reliance on a single LLM judge can introduce model-specific biases. To address this concern, we evaluate system outputs using two additional judges: Gemini-2.5-Pro (a proprietary alternative) and GPT-OSS-20B (an open-weight model). Table 7 presents a comparative analysis of evaluation scores for two deep research systems using the ClueWeb22-B corpus on a sample of 100 queries.

Despite absolute score variance across judges, their per-query assessments align. Correlations are computed against `gpt-4.1-mini`. For both deep research systems, Gemini-2.5-Pro and GPT-OSS-20B both show strong agreement on relevance metrics ($\rho \approx 0.8$) and moderate agreement on faithfulness and quality ($\rho \approx 0.4$–$0.6$). These correlations suggest that while different judges may apply different scoring scales or exhibit distinct biases, the relative ordering of system performance remains consistent.

Table 8: Comparison of deep research systems on the Researchy Questions test set using (i) each system's original commercial search API and (ii) DEEPRESEARCHGYM's FineWeb search API. Scores are judged by `gpt-4.1-mini-2025-04-14`. Systems marked with * are not tailored for long-report generation.

| | Relevance | | | | Faithfulness | | | | Quality | | | |
|---|---|---|---|---|---|---|---|---|---|---|---|---|
| | Commercial | | FineWeb | | Commercial | | FineWeb | | Commercial | | FineWeb | |
| System | KPR | KPC | KPR | KPC | Precision | Recall | Precision | Recall | Clarity | Insight | Clarity | Insight |
| perplexity-sonar-deepsearch | **72.50** | 1.12 | – | – | 55.65 | **99.22** | – | – | **89.50** | **89.26** | – | – |
| gpt4-search-preview | 40.01 | 1.69 | – | – | 57.68 | 56.11 | – | – | 70.13 | 59.13 | – | – |
| GPT-Researcher | 60.61 | 1.52 | 64.47 | 1.54 | 89.11 | 94.29 | 87.42 | 92.41 | 86.37 | 81.52 | 82.90 | 77.73 |
| OpenDeepSearch | 32.92 | 0.97 | 40.73 | 0.86 | 85.86 | 97.78 | 82.7 | 95.19 | 59.20 | 47.04 | 61.25 | 50.01 |
| HuggingFace-DeepSearch | 33.00 | 0.81 | 38.16 | 1.32 | 0.35 | 0.29 | 0.12 | 0.13 | 57.52 | 47.98 | 55.81 | 47.62 |

Finally, our results in Table 2 leverage the ClueWeb22 for search, given its better alignment with the Researchy Questions dataset. In Table 8, we show results for the report-oriented deep research system using DEEPRESEARCHGYM's FineWeb endpoint for search. Results are inline with those previously reported in Table 2, solidifying this endpoint's utility.

## G  QUALITATIVE ANALYSIS: FAILURE MODES

To better understand the limitations of current deep research systems, we perform a qualitative analysis on GPTResearcher, the best-performing open-source system in our benchmark. We selected the 100 queries where it achieved the lowest key-point recall and identified three major failure modes through manual inspection. Below we describe each one, and exemplify with a report (reports are truncated with elipsis due to their considerable length).

Multi-facet coverage gaps account for 78.5% of missing key-points, and occur when a query is broad and requires synthesizing multiple perspectives or roles. GPTResearcher produces a coherent report focused on the most salient interpretation but fails to cover additional sub-facets or long-tail details. For instance, on the query "do the american population wish more autonomy in their work?", the system generated a comprehensive report on workplace autonomy, covering flexibility, remote work, and work-life balance. But it missed gold key points addressing autonomy in other contexts: undocumented workers and immigration policy, physicians' private practice ownership, patient autonomy in medical ethics, and generational narratives about Generation X. Despite producing a seemingly complete answer about workplace autonomy, the report failed to recognize the query's multi-faceted scope:

```
# Report on American Workers' Desire for Greater Autonomy in the Workplace

## Introduction
(...)

## Defining Autonomy and Its Importance
(...)

## Evidence of Desire for More Autonomy Among American Workers

### Survey Data on Autonomy and Flexibility Preferences
(...)

### Autonomy as a Response to Micromanagement
(...)
```

```
### Autonomy and Employee Well-Being
(...)

### Autonomy and Retention
(...)

### Autonomy Across Demographics and Regions
(...)

### Autonomy Versus Flexibility
(...)

## Quantitative Data on Autonomy in the U.S. Workforce
(...)

## Challenges and Considerations in Implementing Autonomy
(...)

## Conclusion and Opinion
(...)
```

Lens mismatches account for 17.9% of missing key-points, and arise when GPTResearcher answers at the wrong level of abstraction or conceptual framing. The report is thematically correct but doesn't align with how the gold key points are structured. For example, when asked "what makes an athlete elite", the system discussed psychological traits, training environments, coaching relationships, and genetic factors. All these are reasonable aspects of elite performance. However, the gold key points were drawn from a source emphasizing neuromuscular mechanisms: brain-muscle communication, motor control centers, and training-induced neural adaptations. Because the report never engaged with this mechanistic level, it failed to capture these key points despite being broadly accurate:

```
# What Makes an Athlete Elite: An In-Depth Analysis
Elite athletes are often admired for their extraordinary (...) that
    collectively contribute to elite athletic performance.
---
## Psychological Characteristics and Mental Skills
### Supreme Concentration and Focus
(...)

### Commitment to Excellence and Motivation
(...)

### Self-Awareness and Emotional Control
(...)

### Optimism and Positive Mindset
(...)

## Environmental and Social Factors
(...)

### The Coach-Athlete Relationship
```

```
(...)

### Support Systems and Training Environment
(...)

### Culture of Excellence
(...)

## Genetic and Biological Influences
```

Finally, domain misinterpretations account for 3.6% of missing key-points, and represent more severe failures where ambiguous terminology leads the system to research the wrong topic entirely. On the query "what role did indians play in the wars for empire?", GPTResearcher interpreted "Indians" as referring to people from India and produced a report on Indian troops in British imperial wars. The gold key points, however, follow a perspective where "Indians" refers to Native Americans and Indigenous peoples, discussing Aztec, Inca, and Maya societies and Indigenous alliances with European powers in eighteenth-century North American conflicts:

```
# The Role of Indians in the Wars for Empire: A Comprehensive Analysis

India's history is deeply intertwined with numerous wars and conflicts (...)

## Historical Context of Indian Warfare and Military Organization
(...)

## Indian Armies in the Medieval and Early Modern Periods
(...)

## Indian Participation in Imperial Wars under British Rule
(...)

## The Indian National Army and Anti-Colonial Struggles
(...)

## Post-Independence Military Engagements
(...)

## Indians in Wars for Empire: Roles and Contributions
(...)
```

# H ADDITIONAL DETAILS ON SEARCH AGENT TRAINING

This appendix provides additional implementation details for the search agent training experiments described in Section 5.

## H.1 DATA SYNTHESIS AND TRAINING CONFIGURATION

Training queries are synthesized following recent approaches (Gao et al., 2025; Shi et al., 2025; Li et al., 2025a), but grounded over ClueWeb22 rather than the live web. Given a root Wikipedia document from ClueWeb22, we prompt an LLM to generate queries targeting information in that document. To introduce multi-hop reasoning, we leverage the document's link structure to identify related entities for constructing

questions requiring synthesis across multiple documents. We also employ entity abstraction (e.g., replacing "Barack Obama" with "44th U.S. President") to encourage generalization beyond surface-level matching. All generated queries undergo rigorous validation via LLM judging to ensure they are (i) answerable, (ii) have a single correct answer, and (iii) require search rather than relying solely on parametric knowledge.

The RL training is conducted for 150 steps, with a batch size of 32 and a group size of 8 for GRPO. At inference time, we use greedy decoding with temperature 0.

## H.2 OBTAINING THE CLUEWEB-TEST EVALUATION SET

Standard benchmarks like GAIA, WebWalkerQA, and HLE are not guaranteed to be covered by ClueWeb22, as the documents supporting ground-truth answers may not exist in the snapshot. To construct a ClueWeb-grounded evaluation set, we filter these benchmarks using a validation agent. This agent uses the same action space as our trained models (search, summarize, answer) but employs Gemini-2.5-Flash as the backbone LLM. We allow the agent to search for up to 30 turns per query. A query is retained in ClueWeb-Test if it satisfies two criteria: (i) the agent achieves pass@5 success, and (ii) the agent performs at least one search action during successful trajectories, confirming that the answer requires information retrieval rather than relying solely on parametric knowledge. This filtering yields a 53-query subset where both training infrastructures have access to the necessary information for meaningful comparison.

