# OpenReview forum: "DeepResearchGym: A Free, Transparent, and Reproducible Evaluation Sandbox for Deep Research"
_ICLR.cc/2026/Conference — Submitted to ICLR 2026_

### Official Review · Reviewer_FX47 · 2025-10-20

**Soundness:** 3
**Presentation:** 3
**Contribution:** 3
**Rating:** 4
**Confidence:** 4

**Summary:**

This paper introduces DeepResearchGym, a benchmark for evaluating deep research systems. The benchmark provides standardized search and fetch APIs to ensure a fair and transparent environment for comparing various systems. Although the benchmark utilizes an 'LLM-as-a-judge' approach, its metrics are validated by human annotation and high inter-annotator agreement. A comparative analysis of two commercial, three white-box, and two baseline deep research systems was conducted using this benchmark.

**Strengths:**

* Provide a transparent and fair environment to compare various deep research systems.
* Include extensive metrics during evaluation

**Weaknesses:**

* Novelty: The current corpus and queries of DeepResearchGym are from existing datasets (ClueWeb22, FineWeb, and Researchy Questions). I am very willing to raise my score if there are any missing innovative details in the methodology.

* A few missing details in the benchmark construction and evaluation. Please see the details in the questions.

**Questions:**

* Line 145: The authors state that low-quality and spam pages were filtered during sampling. Could you clarify if this filtering was part of the original ClueWeb22 dataset's preprocessing, or if an additional filtering step was applied? If it was an additional step, please provide more details on the process.
* The paper mentions that users can switch between corpora (Lines 200-202), yet all evaluations are conducted on either commercial search engines or the ClueWeb22 dataset. The justification for including the FineWeb corpus is unclear. To demonstrate its value and analyze the system's performance across different retrieval corpora, please consider adding experiments based on FineWeb, perhaps in the appendix.
* Figure 1 indicates high efficiency for the search API/retriever in DeepResearchGym. A discussion of the factors contributing to this efficiency would be a valuable addition. Explaining the underlying design choices could further highlight the innovative aspects of the system.
* The evaluation is limited to a small number of systems. While the benchmark aims for fairness, two of the systems evaluated do not use the provided search and fetch APIs, making direct comparison difficult. To provide a more direct and comprehensive comparison, have the authors considered evaluating other generative models within the same standardized pipeline?
* The design of DeepResearchGym is well-suited for efficiency comparisons between different systems (e.g., measuring the number of searches, fetches, or conversational turns). However, the paper currently lacks a discussion on these efficiency metrics. Including an analysis of efficiency would significantly strengthen the evaluation.
* It's always good to include human evaluation as justification for the LLM judges. However, the authors did not discuss the background of the annotators and the instructions for their annotation. Since the task requires expert-level knowledge, it is necessary to have more details regarding human evaluation.

---

> ### Author Response · Authors · 2025-11-21
> **Response**
>
> **W1: Limited novelty**
>
> We acknowledge that our system leverages existing web corpora and benchmarks, making the necessary adaptations. Our primary goal was to establish a reproducible and transparent framework that mitigates dependence on commercial search APIs. The novelty is the availability of the platform itself: to the best of our knowledge, there is no other publicly available large scale search API. Researchers rely on commercial search APIs which are expensive and do not guarantee document retrieval reproducibility across runs. This demand is solidified by the adoption from the community: during the review period, our API received \\~4 million search requests (growing our query log from 12 to 16 million queries).
>
> Within this sandbox, we demonstrated that it can support reliable evaluation by providing controlled retrieval behavior suitable for comparing systems under identical conditions, as detailed in the paper. We also showed that it can function as a practical training environment for agentic search systems, as illustrated in the Global Answer (Section B), where models trained in this setting can generalize to commercial engines at inference time.
>
> We also note that building a large-scale search index involves substantial engineering effort. This included tuning ANN parameters, implementing sharding and load balancing, and ensuring reproducible efficient retrieval across large corpora.
>
> **Q1: The authors state that low-quality and spam pages were filtered during sampling.**
>
> We refer to the filtering procedure described by the ClueWeb22 authors. Their construction pipeline relies on Microsoft Bing’s importance scoring to downweight pages that are unlikely to satisfy user information needs. Our paper does not introduce any additional filtering.
>
> **Q2: Main results are presented for clueweb only.**
>
> We use ClueWeb in the main experiments because Researchy Questions’ relevance judgments are grounded in ClueWeb22. We instantiated this Researchy Questions/ClueWeb22 evaluation pipeline to demonstrate the sandbox’s usability; still researchers may use the FineWeb endpoint if that better suits their needs. For completeness, we conducted an experiment similar to the main table of the paper, but using FineWeb instead of ClueWeb search endpoint on the report-oriented models. The table below show results in-line with those achieved by ClueWeb:
>
> FineWeb searcher; GPT-4.1-mini judge:
> | System | Relevance (KPR) | Relevance (KPC) | Faithfulness (Prec) | Faithfulness (Recall) | Quality (Clarity) | Quality (Insight) |
> |--------|------------------|------------------|-----------------------|-------------------------|--------------------|---------------------|
> | GPTR   | 64.4             | 1.5              | 87.4                  | 92.4                    | 82.9               | 77.7                |
> | ODS   | 40.7             | 0.86              | 82.7                  | 95.2                    | 61.2               | 50.0                |
> | HFDR   | 38.1             | 1.3              | 0.0                   | 0.0                     | 55.8             | 47.6               |         |
>
>
> **Q3: How is efficiency achieved**.
>
> The performance reflects engineering choices. We conducted a systematic hyperparameter search over DiskANN configurations, including graph construction settings and shard layout, and deployed the strongest configuration identified in this process. All design choices were guided by reproducibility, i.e., non-deterministic approaches that would speed up the system were not considered.
>
>
> **Q4: Evaluate more Deep Research Systems with our API**.
>
> We agree that expanding the set of fully controlled systems would strengthen the benchmark; this is planned for future work. For this round, we focused on consolidating the existing setup by testing with different LLM judges, reporting results for both ClueWeb and FineWeb, and showing that the API supports evaluation and model training.
>
> **Q5: Add Deep Research Systems efficiency (num searches, num turns, etc) to the discussion.**.
>
> We ran all models in their default configurations to reflect their real-world behavior, and the prompt-based systems have hard-coded interaction patterns (e.g., fixed numbers of searches or turns). While these could be adjusted through parameter changes, we prioritized evaluating systems as they are typically deployed. Standardizing these patterns would improve fairness in efficiency comparisons and plan to pursue this in future work.

---

> > ### Author Response · Authors · 2025-11-21
> > **Answer (cont)**
> >
> > **Q6: Annotators background and instructions**
> >
> >
> > All annotators were co-authors of the paper with research experience in NLP and IR. The benchmark we used is open-domain, so annotators evaluated questions across the full range of topics without any topic-matching. Please find below the instructions:
> >
> > ---
> >
> > Two reports attempt to answer a research question. Please select which report is better overall, considering the following dimensions:
> >
> > - Does the report provide factually accurate information? Are claims properly supported by evidence?
> >
> > - Are important aspects or perspectives missing?
> >
> > - Does the report draw from credible sources? Does it incorporate multiple perspectives where appropriate?
> >
> > - Is the information well-structured and easy to follow? Does the report effectively synthesize information rather than simply listing facts?

---

> > > ### Comment · Reviewer_FX47 · 2025-11-25
> > >
> > > The authors' comments resolve my previous concerns. I will update my score accordingly.

---

### Official Review · Reviewer_8hFt · 2025-10-28

**Soundness:** 3
**Presentation:** 3
**Contribution:** 3
**Rating:** 6
**Confidence:** 3

**Summary:**

The paper introduces DEEPRESEARCHGYM, an open, reproducible sandbox for evaluating deep research systems that generate long-form, evidence-grounded reports. It provides a free search API over public web corpora (ClueWeb22 A/B and FineWeb), implemented with a dense retriever (MiniCPM-Embedding-Light) and DiskANN, plus endpoints to search and to fetch archived page snapshots for stable, auditable evidence. The authors pair this with a tri-part evaluation protocol on Researchy Questions that scores (i) coverage via Key-Point Recall/Contradiction, (ii) retrieval faithfulness via citation recall/precision, and (iii) report quality (clarity, insight) using LLM-as-judge prompts. Empirically, systems keep similar rankings when swapping commercial search for this API, and the API achieves sub-0.5s median latency, suggesting it is a practical, research-grade substitute for proprietary search backends.

**Strengths:**

1. This paper proposes DeepResearchGym, an open-source benchmarking framework specifically designed to enable transparent and reproducible evaluation of deep research systems. Being free and open-source makes this work a valuable resource to the community.

2. Empirical evaluations show that the system achieves strong retrieval quality with minimal loss from approximate search, as well as maintaining response times below those attained by commercial APIs.

3. The paper is well-written and easy to follow.

**Weaknesses:**

1. Although DeepResearchGym helps the reproduction of deep research systems with a higher response speed and lower cost, using static corpora may under-serve very time-sensitive queries.

2. Despite showing high correlation with human evaluation, the empirical results only on Researchy Questions using GPT-4.1 as a judge might not comprehensively and robustly reflect the system's actual performance to serve as a replacement of search engine. Additionally, more fine-grained comparison between the proposed API vs different commercial API would be much appreciated. What's the fundamental difference between different search APIs, and how would that affect the performance of deep research systems?

**Questions:**

See weaknesses

---

> ### Author Response · Authors · 2025-11-21
> **Response**
>
> We appreciate the reviewer’s feedback. Below we address each weakness and answer each question individually.
>
> **W1: Static vs Live Web**
>
> We concur that there is an inherent trade-off between using static web corpora for reproducibility, and the inability to evaluate time-sensitive topics. This was a deliberate methodological choice when designing the framework, as our primary goal is to provide a transparent and reproducible search sandbox that enables controlled benchmarking, which was previously infeasible due to the reliance on dynamic commercial APIs.
>
> Nonetheless, we acknowledge the lack of freshness remains a limitation. We plan to include other corpora, such as updated FineWeb crawls when they become available and can be stably integrated. These additions, while still representing static snapshots at any given time, will enhance the breadth and recency of the content available for reproducible research, balancing the need for stability with the desire for more current information.
>
> We also note that handling time-sensitive topics is a very relevant challenge for deep research systems that goes beyond evaluation. Timestamps are not reliably associated as metadata for contents on the Web, and thus the appropriate handling of temporal information needs relies on interpreting and reasoning over temporal expressions in the textual contents of both user queries and retrieval contents. These aspects remain as interesting for exploration in the context of deep research systems.
>
> Finally, we do not aim to replace live search. Live systems remain appropriate for tasks and benchmarks that require real-time information or deployment-oriented evaluations. Our goal is to complement it by providing a stable environment that supports controlled experimentation and that can generalize aspects of real-world search behavior in a way that is suitable for research.
>
> **W2.1: Comprehensiveness of evaluation**
>
> To mitigate concerns about the usage of a single model, we conducted an additional experiment using Gemini-2.5-Pro and GPT-OSS-20B (open weight) as the judges. These results will shortly be added to the manuscript pdf. The tables below presents a comparative analysis of evaluation scores for two deep research systems using the ClueWeb22-B corpus for search on a sample of 100 queries:
>
> GPT-4.1-mini
> | System | Relevance (KPR) | Relevance (KPC) | Faithfulness (Prec) | Faithfulness (Recall) | Quality (Clarity) | Quality (Insight) |
> |--------|------------------|------------------|-----------------------|-------------------------|--------------------|---------------------|
> | GPTR   | 67.4             | 1.5              | 87.0                  | 89.3                    | 82.1               | 76.5                |
> | HFDR   | 38.4             | 0.9              | 0.0                   | 0.0                     | 56.9               | 51.1                |
>
> Gemini-2.5-pro
> | System | Relevance (KPR) | Relevance (KPC) | Faithfulness (Prec) | Faithfulness (Recall) | Quality (Clarity) | Quality (Insight) |
> |--------|------------------|------------------|-----------------------|-------------------------|--------------------|---------------------|
> | GPTR   | 60.2             | 4.8              | 84.4                  | 92.7                    | 50.8               | 57.6                |
> | HFDR   | 43.2             | 2.7              | 0.0                   | 0.0                     | 31.9               | 31.7                |
>
> GPT-OSS-20B
> | System | Relevance (KPR) | Relevance (KPC) | Faithfulness (Prec) | Faithfulness (Recall) | Quality (Clarity) | Quality (Insight) |
> |--------|------------------|------------------|-----------------------|-------------------------|--------------------|---------------------|
> | GPTR   | 59.5             | 2.8              | 78.8                  | 63.3                    | 69.7               | 45.9                |
> | HFDR   | 38.2             | 1.7              | 0.0                   | 0.0                     | 45.2               | 32.2                |
>
> While absolute scores vary across judges, their per-query assessments align. Correlations are computed against gpt-4.1-mini. For both deep research systems, Gemini-2.5-Pro and GPT-OSS-20B both show strong agreement on relevance (rho about 0.8) and moderate agreement on faithfulness and quality (rho about 0.4–0.6).
>
> A revised version of the paper can discuss the potential benefits of using state-of-the-art LLM as judges, whose performance may evolve, versus using a fixed local LLM for judging, which would be beneficial for long-term reproducibility.
>
> To motivate the potential of our search engine as a complement for commercial options, we highlight the experiment summarized in the Global Response, shows that DeepResearchGym can serve as a stable and cost-efficient training environment for agentic search systems.

---

> ### Author Response · Authors · 2025-11-21
> **Response (cont)**
>
> **W2.2: Other Search APIs:**
>
> Most commercial search providers used in open-source deep research systems crawl over the same underlying Google results ([1,2]), so they tend to return similar document sets. We therefore decided to keep each system’s original commercial API configuration, which keeps the setup consistent with how these systems were originally implemented.
>
> [1] Alzubi et al. (2025) Open Deep Search: Democratizing Search with Open-source Reasoning Agents.
> [2] Gao et al. (2025) Beyond Ten Turns: Unlocking Long-Horizon Agentic Search with Large-Scale Asynchronous RL

---

> > ### Comment · Reviewer_8hFt · 2025-11-25
> > **Response to Authors' Rebuttal**
> >
> > Thanks for the detailed response and clarification. Overall, it's a good and timely work that helps build open-source deep research systems. I'll stand for my score.

---

### Official Review · Reviewer_8rKV · 2025-10-28

**Soundness:** 3
**Presentation:** 3
**Contribution:** 3
**Rating:** 6
**Confidence:** 3

**Summary:**

This paper introduces **DeepResearchGym**, an open-source benchmarking framework for evaluating deep research systems. The framework addresses critical reproducibility and transparency issues in current evaluation practices that rely on proprietary, dynamic commercial search APIs. The main contributions are: (1) A **free search API** built on large-scale public web corpora (ClueWeb22 and FineWeb) with dense retrieval (MiniCPM-Embedding) and approximate nearest neighbor search (DiskANN), achieving lower latency than commercial alternatives while ensuring stable rankings; (2) An **evaluation protocol** extending the Researchy Questions benchmark with LLM-as-a-judge metrics measuring report relevance (Key Point Recall/Contradiction), retrieval faithfulness (citation precision/recall), and report quality (clarity/insightfulness); (3) **Empirical validation** showing systems maintain comparable performance when switching from commercial to DeepResearchGym APIs, with human evaluation confirming metric reliability ($\kappa = 0.72$-$0.89$).

**Strengths:**

1. **Addresses critical reproducibility gap**: The field urgently needs standardized benchmarking infrastructure. The paper directly tackles cost, transparency, and reproducibility issues with commercial APIs.
2. **Comprehensive multi-dimensional evaluation**: The three-faceted framework (relevance, faithfulness, quality) captures different aspects of report generation quality, going beyond simple surface-form metrics.
3. **Nuanced analysis**: Query-level correlation analysis (Figure 2) and query log analysis (Appendix A) provide insights beyond system-level aggregates.

**Weaknesses:**

1. **Heavy reliance on a single judge model without robustness analysis**: All automatic evaluations use GPT-4.1-mini exclusively as the judge, with no ablation studies using alternative models such as GPT-4o, Claude Sonnet, or open-source alternatives. This creates potential concerns about judge-specific biases, particularly since some evaluated systems (like OpenAI's deep research) are based on GPT models.
2. **Insufficient examination of static corpus limitations and ground-truth quality**: While the paper demonstrates that systems maintain performance when switching to static corpora, it does not analyze when this approach is adequate versus problematic. There is no quantitative assessment of which query types are temporally sensitive and might suffer from the 2022-2024 snapshot. Additionally, the evaluation assumes that clicked documents from search logs constitute comprehensive ground truth, but this assumption is never validated. Users may click tangential documents or miss important sources, yet the paper provides no analysis of click quality or relevance distribution.

**Questions:**

1. What is the correlation between report length and KPR? Are higher-scoring systems simply more verbose?
2. What are the most common failure modes? Can you provide qualitative analysis of queries where systems perform poorly vs. well?

---

> ### Author Response · Authors · 2025-11-21
> **Response**
>
> We appreciate the reviewer’s feedback. Below we address each weakness and answer each question individually.
>
>
> **W1: Single LLM judge**
> We agree with the reviewer that reliance on LLM-as-a-judge for evaluation can introduce biases. This is a common and acknowledged challenge in the evaluation of long-form generation. Our reliance on gpt-4.1-mini for the LLM-as-a-judge protocol was based on efficiency, its state-of-the-art performance, and availability at the time of our experiments.
>
> To mitigate concerns about the usage of a single proprietary model, we conducted an additional experiment using Gemini-2.5-Pro and GPT-OSS-20B (open weight) as the judges. These results will shortly be added to the manuscript pdf. The tables below presents a comparative analysis of evaluation scores for two deep research systems  using the ClueWeb22-B corpus for search on a sample of 100 queries:
>
> GPT-4.1-mini
> | System | Relevance (KPR) | Relevance (KPC) | Faithfulness (Prec) | Faithfulness (Recall) | Quality (Clarity) | Quality (Insight) |
> |--------|------------------|------------------|-----------------------|-------------------------|--------------------|---------------------|
> | GPTR   | 67.4             | 1.5              | 87.0                  | 89.3                    | 82.1               | 76.5                |
> | HFDR   | 38.4             | 0.9              | 0.0                   | 0.0                     | 56.9               | 51.1                |
>
> Gemini-2.5-pro
> | System | Relevance (KPR) | Relevance (KPC) | Faithfulness (Prec) | Faithfulness (Recall) | Quality (Clarity) | Quality (Insight) |
> |--------|------------------|------------------|-----------------------|-------------------------|--------------------|---------------------|
> | GPTR   | 60.2             | 4.8              | 84.4                  | 92.7                    | 50.8               | 57.6                |
> | HFDR   | 43.2             | 2.7              | 0.0                   | 0.0                     | 31.9               | 31.7                |
>
> GPT-OSS-20B
> | System | Relevance (KPR) | Relevance (KPC) | Faithfulness (Prec) | Faithfulness (Recall) | Quality (Clarity) | Quality (Insight) |
> |--------|------------------|------------------|-----------------------|-------------------------|--------------------|---------------------|
> | GPTR   | 59.5             | 2.8              | 78.8                  | 63.3                    | 69.7               | 45.9                |
> | HFDR   | 38.2             | 1.7              | 0.0                   | 0.0                     | 45.2               | 32.2                |
>
> While absolute scores vary across judges, their per-query assessments align. Correlations are computed against gpt-4.1-mini. For both deep research systems, Gemini-2.5-Pro and GPT-OSS-20B both show strong agreement on relevance (rho about 0.8) and moderate agreement on faithfulness and quality (rho about 0.4–0.6).
>
> A revised version of the paper can discuss the potential benefits of using state-of-the-art LLM as judges, whose performance may evolve, versus using a fixed local LLM for judging, which would be beneficial for long-term reproducibility.
>
> **W2: Static search limitations; Ground-thruth quality**
>
> W2.1 - “quantitative assessment of which query types are temporally sensitive and might suffer from the 2022-2024 snapshot”:
>
> Since the Researchy Questions dataset is grounded on ClueWeb (i.e., the ground-truth documents are in the Clueweb22 collection), none of the queries we used for the experiments lack coverage. To improve coverage, we included ClueWeb-A instead of just ClueWeb-B, which contains approximatelly 1 billion documents. FineWeb is also included covering documents up to late 2024, and we remain commited to support more recent FineWeb crawls to maintain document freshness, all while providing separate endpoint to provide reproducibility.
>
> W2.2 - “evaluation assumes that clicked documents from search logs constitute comprehensive ground truth, but this assumption is never validated”:
>
>
> The Researchy Questions benchmark does not explicitly validate clicked documents as ground truth, but this follows standard IR practices. Aggregated clicks from multiple users provide reliable implicit relevance signals, helping to reduce noise from individual errors [1]. MS MARCO similarly uses clicked passages from Bing logs as positive examples [2]. Researchy Questions further filters for high-frequency queries with multiple distinct clicked URLs to strengthen signal quality and mitigate outliers.
>
> [1] Joachims et al. (2005) Accurately Interpreting Clickthrough Data as Implicit Feedback.
> [2] Bajaj et al. (2016) MS MARCO: A Human Generated MAchine Reading COmprehension

---

> > ### Author Response · Authors · 2025-11-21
> > **Response (cont)**
> >
> > **Q1: correlation between KPR and report length**
> >
> > KPR correlates with word count at r=0.63, a moderate relationship. Longer answers often include more key points, but many verbose outputs still miss essential evidence.
> >
> >
> > **Q2: Failure modes qualitative analysis**
> >
> > We are currently conducting a qualitative analysis and will update shortly. Preliminary inspection shows two dominant failures: weak retrieval that provides insufficient evidence and strong retrieval where the model hallucinates or misinterprets content. A detailed analysis can be added in a revision.

---

> > > ### Comment · Reviewer_8rKV · 2025-11-26
> > >
> > > Thank you for the thoughtful rebuttal. Your additional experiments substantially address my primary concerns. The remaining ones:
> > >
> > > ## W2.1: Temporal Sensitivity
> > >
> > > Your response doesn't fully address my concern. The issue isn't whether documents exist in the corpus, but rather **when queries require temporally recent information**. For example, "Why have used car prices increased?" asked in 2025 should discuss 2025 tariffs, but ClueWeb22 is from 2022. I recommend adding:
> > > - A brief taxonomy of query temporal sensitivity (e.g., "current events" vs. "conceptual")
> > > - Discussion of when static corpora are appropriate vs. limiting
> > >
> > > ## Q2: Failure Modes
> > >
> > > "Currently conducting" is insufficient. Even 10-15 examples with brief categorization would be valuable. The two preliminary failure modes (weak retrieval vs. hallucination) are interesting. Please quantify these and include in revision.
> > >
> > > Overall, it's a good and timely contribution, and I keep my original rating.

---

> > > > ### Author Response · Authors · 2025-12-03
> > > > **Response**
> > > >
> > > > Thank you for the follow-up. We address the two remaining points below:
> > > >
> > > > **W2.1: Temporal sensitivity**
> > > > We have now addressed this directly in the manuscript, categorizing queries into temporally stable and temporally evolving as per your suggestion. We quantify each of those types, and discuss static corpora limitations. This can be found in Appendix B.
> > > >
> > > > **Q2: Failure modes**
> > > > We added a new Appendix G, where we identified and quantified error modes. Each error mode is exemplified with a report. We did a more robust analysis than the previous high level inspection, focusing on instances where the best performing model (GPTResearcher) achieved low key-point recall. We identified three error modes:
> > > >
> > > > - Multi facet coverage gaps: the system focuses on the most salient interpretation of a broad query and misses secondary facets or long tail details.
> > > >
> > > > - Lens mismatches: the system answers at a thematically valid level but does not capture the depth required by the gold key points.
> > > >
> > > > - Domain misinterpretations: the system misreads ambiguous terminology and generates a report about the wrong topic.

---

### Official Review · Reviewer_Rq91 · 2025-11-01

**Soundness:** 2
**Presentation:** 2
**Contribution:** 2
**Rating:** 4
**Confidence:** 4

**Summary:**

This work introduces DeepResearchGym, an open-source sandbox for deep research agents. The sandbox features a search API implementation that is reproducible to facilitate future research and agent development. Based on the sandbox, the authors further evaluated several state-of-the-art deep research agents and presents insight into their performance along different axes, including relevance, faithfulness, and quality. Results show that agents can achieve comparable performance when using DeepResearchGym compared to using commercial search APIs.

**Strengths:**

1. The paper contributes an open-source sandbox that can potentially facilitate reproducible research in deep research agents.
2. The authors deliver some insight into fine-grained performance of popular deep research agents.

**Weaknesses:**

1. line 154-155: While the authors use recent data sources to construct the sandbox, it is unclear from the paper how to keep the sandbox "up-to-date".
2. The contribution beyond the sandbox is limited, as the authors mostly follow existing work in their evaluation protocol.
3. While I understand the importance of having a reproducible environment for benchmarking deep research agents, I fail to see the discussion on the actual benefit from this paper. What are the evaluation nuances that DeepResearchGym helps to capture which are infeasible with time-varying search APIs?
4. In section 4.3, the authors use the variability observed in query-level analysis to justify the importance of using a standard retrieval API. I think the logic is flawed. Doesn't this show that the evaluation metrics, which heavily relies on textual overlap, are sensitive the search APIs used?
5. For section 4.4, I don't see the meaning of this pairwise human evaluation when the main results in the paper are pointwise. Why not letting human evaluators follow the same protocol as LLM-as-a-judge and compute score correlations?

**Questions:**

1. line 144, how do you define "low quality"?
2. line 178-179, how do you define "minimal loss"?
2. Section 4.2, why are subjective report quality metrics (Clarity, Insight) comparable to objective information coverage metrics (KPR)?

---

> ### Author Response · Authors · 2025-11-21
> **Response**
>
> We appreciate the reviewer’s feedback. Below we address each weakness and answer each question individually.
>
> **W1: Unclear how to keep the sandbox updated**
> For the endpoint we make publicly available, we are committed to regular updates with more recent FineWeb crawls. We will support versioned endpoints for each new crawl, ensuring reproducibility and up-to-date collections. Furthermore, researchers can leverage the open-source code to locally index whichever corpora they need for experimentation.
>
> **W2: Limited novelty**
> We acknowledge that our system leverages existing web corpora and benchmarks, making the necessary adaptations. Our primary goal was to establish a reproducible and transparent framework that mitigates dependence on commercial search APIs. The novelty is the availability of the platform itself, as to the best of our knowledge, there is no other publicly available large scale search API. Researchers rely on commercial search APIs which are expensive and do not guarantee document retrieval reproducibility across runs. This is a key motivation: the scientific need for a controlled research environment that complements Google SERP crawling. Scientific work requires reproducibility, and progress cannot rely solely on commercial search engines that change behavior and ranking policies without notice. DeepResearchGym is designed to fill this gap by supplying an open and stable environment that supports replicable experiments. This demand is solidified by the adoption from the community: during the review period, our API received \\~4 million search requests (growing our query log from 12 to 16 million queries).
> To reinforce the practical value of this contribution, we highlight the experiment summarized in the Global Answer (Section B), showing that DeepResearchGym can serve as a stable and cost-efficient training environment for agentic search systems.
>
> **W3: Unclear benefits of DRGym API VS time-varying commercial APIs**
>
> Our API ensures retrieval reproducibility, full transparency of the document collection and free access for large-scale experimentation. It also avoids query leakage concerns associated with commercial search providers, which strengthens privacy guarantees for researchers.
>
>
>
> **W4: Variability in query level analysis entails that the evaluation metrics are sensitive to search APIs.**.
>
> The query-level analysis shows that running deep research systems with different search backends can lead to unwanted variations for the same question. We argue that using a single transparent and stable search option, as provided in our framework, offers a more consistent basis for benchmarking.
>
>
> **W5: Human evaluation is pairwise while main tables present point-wise results.**.
>
> A fully pointwise assessment would require extensive key-point extraction and URL-level verification for every query, which is not feasible at scale. Instead, we opted for a more objective analysis, common in arena-style papers, where annotators would choose the best report for a given query.
>
> An LLM can also perform the same pairwise task as an additional evaluation axis. We add here an example of one such evaluation: for the same query pool that was judged by humans, gpt-4.1-mini is used to run the A-B testing. The matrix below shows the LLM-perceived win rate for each model against others:
>
> | X ↓ vs Y → | GPTR | ODS | HFDS | O1 | R1 |
> |---------------------|-----|-----|------|----|----|
> | GPTR                 | -    | 100% | 100% | 100% | 100% |
> | ODS                 | 0%   | -  | 58%  | 78% | 100% |
> | HFDS                | 0%   | 42%  |-   | 72% | 100% |
> | O1                  | 0%   | 22%  | 28%  | -  | 86% |
> | R1                  | 0%   | 0%   | 0%   | 14%  |- |
>
> The obtained model ranking is consistent with that achieved by other metrics in Table 2 of the paper. Moreover, individual human-llm pairwise agreement is high, with Cohen’s kappa ranging from 0.75 to 0.83.

---

> > ### Author Response · Authors · 2025-11-21
> > **Response (cont)**
> >
> > **Q1: Line 144, how do you define "low quality"?**
> > We refer to the filtering procedure described by the ClueWeb22 authors. Their construction pipeline relies on Microsoft Bing’s importance scoring to downweight pages that are unlikely to satisfy user information needs. Our paper does not introduce any additional filtering.
> >
> > **Q2: Line 178-179, how do you define "minimal loss"?**
> > We refer to the empirical gap between approximate and exact retrieval when using the ANN index. Table 1 shows that MRR remains stable as L increases, which indicates that the rank of the relevant document is preserved. In addition, ANN R@10 at the lowest configuration reaches 90 percent. Together, these two signals show that ANN introduces only a small deviation from the exact ranking.
> >
> > **Q3: why are subjective report quality metrics (Clarity, Insight) comparable to objective information coverage metrics (KPR)**:
> >
> > The comparison between subjective report quality metrics and objective information coverage metrics characterizes the relative difficulty of the two evaluation dimensions. Clarity and Insight capture surface-level qualities, so systems that produce fluent text can score highly on these metrics even when the underlying content coverage is incomplete. In contrast, KPR measures whether key pieces of evidence are retrieved and incorporated. As a result, we observe that KPR tends to be harder to saturate.

---

### Author Response · Authors · 2025-11-21
**Global Response**

We thank the reviewers for their feedback. In individual answers, we have addressed concerns raised by each reviewer, including clarifications on filtering, evaluation protocols, static versus live corpora, judge model robustness, efficiency, and novelty. We added requested experiments, including evaluation using the FineWeb search corpus, alternative LLM judges to confirm robustness, and additional correlation analysis.

**A- Summary of Contribution**
We introduce a sandbox for reproducible research of deepresearch systems, including  a free and stable search API intended to compliment commercial search services. We illustrate its utility by building a full evaluation protocol for deep research systems grounded in ClueWeb22, which provides a controlled environment for measuring search quality and end-to-end system performance. We aim for the platform to be flexible enough to support multiple research needs, so, besides ClueWeb22, we provide searching FineWeb for researchers who need more recent crawl. This setup broadens applicability, enables comparisons across corpora, and supports diverse experimental needs while maintaining full reproducibility.

**B- DeepResearchGym for Model Training**

We take this global answer to further demonstrate the utility of DeepResearchGym with another use-case beyond evaluation. Specifically, we explored its use as a training environment for agentic search systems. Training agents with commercial search APIs is costly, often requiring hundreds of thousands of queries across multiple trajectories. For example, a modest reinforcement learning setup with 10 thousand queries, 16 trajectories per rollout, and 4 searches per trajectory will cost up to 640\\$ with Serper, and 5000\\$ with Tavily. Note that this is for a single training run, with prices quickly piling up under experimental setups that require multiple training runs. In turn, leveraging our API is free, while maintaining latencies similar to the commercial approaches as described in the paper.

We start by synthesizing a training dataset grounded in ClueWeb22. Queries were generated from Wikipedia subset of ClueWeb22, following state-of-the-art synthetization approaches such as ASearcher’s [1], and were designed to include multi-hop reasoning and entity abstraction to encourage generalization beyond surface-level patterns.

We trained two agents to evaluate DeepResearchGym as a training environment. Both agents follow the same simple design [2]: backbone is Qwen3-1.7B, and the model’s action space is search, answer, or summarize context. The first agent was trained on our synthetic dataset (5 thousand queries) grounded in ClueWeb22 with access only to the DeepResearchGym search API. The second agent was trained on a state-of-the-art synthetic dataset [3] (5 thousand queries) with access to a commercial search API. Both agents were then evaluated on standard short-answer benchmarks using commercial search. This choice avoids coverage gaps in ClueWeb22 that can affect open-web evaluation and provides a direct test of generalization. The setup measures whether an agent trained within a confined and fully reproducible environment can transfer its learned search policy to a commercial engine at inference time.

The table below presents success rates on GAIA, WebWalkerQA, and HLE, computed according to each dataset's guidelines. The agent trained with DeepResearchGym achieves competitive performance despite never accessing commercial search during training, matching or exceeding the commercial-trained baseline on GAIA and HLE despite showing lower performance on WebWalkerQA. These results demonstrate that DeepResearchGym provides a cost-effective and reproducible training environment where learned search strategies can transfer across both search engines and evaluation distributions. We will shortly provide a new version of the manuscript pdf, with these experiments included in the main body.

| Method                                  | Search Train | Search Infer | GAIA | WebWalker | HLE |
|-----------------------------------------|--------------|--------------|------|-----------|-----|
| **Before RL**                           |              |              |      |           |     |
| Qwen3-1.7B [1]                        | None         | Serper       | 8.7  | 19.1      | 6.2 |
| **After RL**                            |              |              |      |           |     |
| Qwen3-1.7B + RL (Open Web)              | Serper       | Serper       | 18.4 | 35.4      | 6.9 |
| Qwen3-1.7B + RL (DeepResearchGym)       | ClueWeb      | Serper       | 20.3 | 29.7      | 7.0 |


[1] Jin et al. (2025) Beneficial Reasoning Behaviors in Agentic Search and Effective Post-training to Obtain Them.
[2] Gao et al. (2025) Beyond Ten Turns: Unlocking Long-Horizon Agentic Search with Large-Scale Asynchronous RL.
[3] Li et al. (2025) Chain-of-Agents: End-to-End Agent Foundation Models via Multi-Agent Distillation and Agentic RL.

---

### Author Response · Authors · 2025-12-03
**Global Response 2**

We thank the reviewers for their feedback during the rebuttal period. We take this global answer to summarize the discussion, and update with the most recent changes to the manuscript PDF.

## Reviewer positioning summary


During the discussion period:


- Reviewer 8rKV maintained the rating and stated "it's a good and timely contribution". The reviewer still had two comments after initial discussion, which we now address below in this response.
- Reviewer 8hFt maintained the rating and confirmed "it's a good and timely work that helps build open-source deep research systems".
- Reviewer FX47 raised the rating, noting our responses "resolve previous concerns".
- Reviewer Rq91 did not have the chance to interact with our rebuttal.

## Contributions and strengths as highlighted by reviewers

- **Infrastructure gap addressed (8rKV, 8hFt, FX47):** The field needs standardized and reproducible infrastructure. Our free open-source search API directly tackles cost, transparency, and reproducibility issues with commercial APIs.
- **Comprehensive evaluation framework (8rKV, FX47):** Multi-dimensional metrics capture different aspects of report generation beyond surface-form measures.
- **Strong empirical validation (8hFt, FX47, Rq91):** Provided fine-grained insights into performance of popular deep research systems, showing that systems maintain comparable performance when switching from commercial to our API

## Addressing reviewer’s concerns

We addressed each of the reviewers' questions individually. Below we list a summary, also reflecting changes to the manuscript:

### Judge model robustness (8rKV, 8hFt)
- **Concern:** Work presents results leveraging a single judge model (GPT-4.1-mini).
- **Response:** We conducted experiments with Gemini-2.5-Pro and GPT-OSS-20B showing strong per-query correlation.
- **Paper update:** Appendix F now includes multi-judge analysis and discusses tradeoffs between state-of-the-art versus fixed judges for long-term reproducibility.



### Static corpus limitations (8rKV, 8hFt)
- **Concern:** Time-sensitive queries may be under-served by 2022-2024 snapshots.
- **Response:** This reflects a deliberate design choice prioritizing reproducibility. We plan to support updated FineWeb crawls via versioned endpoints.
- **Paper update:** Appendix B now includes temporal sensitivity analysis, taxonomy of query types, and discussion of when static corpora are appropriate versus limiting.

### FineWeb Inclusion (FX47)
- **Concern:** Results are limited to ClueWeb22; FineWeb inclusion not justified.
- **Response:** Researchy Questions judgments are grounded in ClueWeb22, making it a natural search corpus for main experiments. We conducted additional experiments using FineWeb showing consistent results (GPTResearcher: 64.4% KPR with FineWeb vs. 67.4% with ClueWeb).
- **Paper update:** Appendix F presents complete FineWeb results for all report-oriented systems.

### Novelty and contribution scope (Rq91, FX47)
- **Concern:** Limited novelty since corpus and queries come from existing datasets.
- **Response:** The novelty lies in the infrastructure availability itself. To our knowledge, no other publicly available large-scale search API exists for research. To further demonstrate the usefulness of the contribution, we added experiments showing that our search API can be used as a cost-effective training environment, where agents trained on our API generalize to commercial search at inference time.
- **Paper update:** Section 5 now includes training sandbox experiments showing DeepResearchGym enables cost-effective RL training.

The need for our solution is highlighted by strong adoption, with the search endpoint receiving a total of 16 million requests by the time of writing. This level of access illustrates the demand for open and reproducible tools in deep research systems. Furthermore, by providing a free alternative to commercial search APIs which would be costly at this scale, our platform supports broader participation and democratizes research in this area. We hope this summary assists the AC, and we thank all involved for their engagement and constructive input throughout the review cycle.

---

### Meta-Review · Area_Chair_8DxQ · 2026-01-08

**Summary:**

The paper introduces DeepResearchGym, a benchmarking framework and open-source sandbox for evaluating deep research agents. It provides a free, reproducible search API based on public corpora (ClueWeb22 and FineWeb) to replace dynamic commercial APIs. The authors also propose an evaluation protocol using the Researchy Questions dataset with LLM-as-a-judge metrics for relevance, faithfulness, and report quality.

**Reviewer Concerns:**

**Addressed:**

**Judge Model Bias:** Reviewers 8rKV and 8hFt raised concerns about relying solely on GPT-4o-mini as a judge. The authors added experiments using Gemini-2.5-Pro and GPT-OSS-20B, demonstrating that relative rankings of agents remained consistent across judges.

**Corpus Coverage:** Reviewer FX47 questioned the exclusion of FineWeb results in the main text. The authors provided these results in the rebuttal.

**Utility/Baselines:** Reviewer Rq91 questioned the benefit over commercial APIs. The authors demonstrated the sandbox's utility as a cost-effective training environment where agents trained on the static API generalized to commercial search tools.



**Outstanding:**

**Static Corpus Limitations:** A major, persistent concern from Reviewers 8rKV and 8hFt is that using static snapshots (2022-2024) fundamentally limits the evaluation of "Deep Research" agents, which often require up-to-date information. While the authors argue this is a trade-off for reproducibility , Reviewer 8rKV noted that the rebuttal did not fully address the taxonomy of when this limitation invalidates the benchmark (e.g., current events).


**Novelty:** Reviewers Rq91 and FX47 noted that the paper combines existing datasets and metrics without introducing significant algorithmic novelty. The contribution is primarily infrastructural.

**Reviewer Scores:**

**Reviewer Rq91 (Current: 4):** will be 4. The authors addressed the "benefit" question with the training experiments.

**Reviewer 8rKV (Current: 6):** will be 6. The reviewer explicitly maintained their score after the rebuttal, stating the contribution is "good and timely" but noting the outstanding concerns about temporal sensitivity.


**Reviewer 8hFt (Current: 6):** will be 6. This reviewer also explicitly maintained their score, acknowledging the utility for open-source systems but keeping the rating due to the static corpus limitation.

**Reviewer FX47 (Current: 4):** will be 6. The reviewer stated they would "update my score accordingly" following the resolution of their concerns.

---

### Decision · Program_Chairs · 2026-01-26

Reject